# Influence of Nonlinear Dynamics Behavior of the Roller Follower on the Contact Stress of Polydyne Cam Profile

**Louay S. Yousuf**

Department of Mechanical Engineering, San Diego State University, 5500 Campanile Drive, San Diego, CA 92182-1323, USA; louaysabah79@yahoo.com

**Abstract:** The effect of the cam profile on the nonlinear dynamics phenomenon of the follower is studied at three involutes' profiles for the cam. The value of the Lyapunov exponent parameter is calculated at different internal distances of the follower guide from inside and at different cam speeds. The effect of the Lyapunov exponent value on the contact stress is studied based on the clearance between the follower and its guides. The contact between the cam and the square grooving key and between the cam and the follower has been taken into consideration at different locations. The finite element method is used to calculate the contact stress numerically using the SolidWorks program. The nonlinear response of the follower is calculated analytically using the Newton–Euler equations of rigid body dynamics of translation and rotation motions while the follower position is tracked experimentally using a high-speed 3-D camera device. The contact stress is checked and verified using photo-elastic apparatus. The data of the contact stress against time are used in the Wolf algorithm code to extract the value of the largest Lyapunov exponent. A phase-plane diagram is drawn at different cam speeds and different internal distances of the follower guide from the inside. The periodic and non-periodic motions of the contact stress are examined at different contact locations for the follower with the cam profile. The higher the value of contact stress, the higher the value of the largest Lyapunov exponent parameter and the higher the value of bending deflection.

**Keywords:** nonlinear dynamics; nonlinear response; Lyapunov exponent; phase-plane diagram





## 1. Introduction

The main cause of high contact stress on the cam profile is the clearance between the follower and its guide and when the follower moves with three degrees of freedom at high cam speeds. Many researchers have studied the nonlinear dynamics phenomenon in the cam-follower system but the effect of the nonlinear dynamics phenomenon on the contact stress between the cam and the follower at high speeds has not been discussed yet. The proposed cam-follower system can be found in textile machines, automatic lathe machines, hydraulic systems, and internal combustion engines. There are some factors that affect the contact stress between the cam and follower: (a) when the cam spins at non-constant speeds in the presence of follower offset; (b) using the cam with a small radius of curvature for the nose or when the cam has a discontinuities profile; (c) due to the impact and the friction between the follower and its guide and between the cam and the follower. Yousuf and Marghitu studied the influence of flank curvature of the cam profile on the nonlinear dynamics behavior of the roller follower at different cam speeds and different internal distances of the follower guide from inside [1]. They found the nonlinear response of the roller follower at different contact conditions. Moreover, Flores improved the cam profile by using maximum pressure angle and minimum radius of curvature for a disk cam with eccentric translating roller follower [2,3]. The bifurcation diagram has been built to describe the dynamic motion of the follower due to high contact stress based on the difference in angular velocity between the cam and the follower [4]. Ouyang et al. employed a modified trapezoidal curve for the follower motion to improve the dynamic motion by using the

strategy of a genetic algorithm. They used a single objective function in ADAMS software to optimize the cam profile in order to prevent follower detachment [5]. The optimum size of the cam profile is found by Dgeddou et al. based on the radius of the base circle of the cam, radius of the roller follower, and follower's offset [6]. Nguyen et al. reduced the error in cam profile in which the nonlinear equation has been solved using the Lagrangian finite element method. They concluded that the accuracy of the cam profile is increased with the increasing number of elements using finite element method [7]. On the other hand, Lassaad used a nonlinear model of Hertzian contact with eight degrees of freedom to reduce the error in the cam profile [8]. Yang et al. used a tangential slip theory to show that the cam and the follower stayed in permanent contact at low speeds of the cam [9]. The aim of this article is to suppress the nonlinear dynamics phenomenon of the follower and to reduce the contact stress between the cam and the follower. Multi degrees of freedom (spring-damper-mass) systems on the follower stem are used at different cam speeds and different internal distances of the follower guide from inside.

## 2. Analytic Calculations of the Nonlinear Response of Vibration

The Newton–Euler equations of rigid body dynamics of translation and rotation motions were applied on the geometry of the follower with multi degrees of freedom (spring-damper-mass) systems. The vibration of the follower stem inside its guides was described by the horizontal spring-damper system in which the spring stiffness is (K1 = K2 = K3 = K4) and the damping coefficient is (C1 = C2 = C3 = C4). The vibration of the follower stem in the vertical direction was described by the spring stiffness (K5) and with the damping coefficient (C5). Figure 1 shows the vibration of the follower stem in the horizontal and vertical directions. The vibration in the cam-follower system was assumed to be multi degrees of freedom [10]. The Newton equation of rigid body dynamics when the follower translates in the vertical direction is:

$$\sum \text{Forces} = m\ddot{x}_2$$

$$P_C - K_5(X_2 - T\theta) - C_5\left(\dot{X}_2 - T\dot{\theta}\right) = m\ddot{x}_2 \tag{1}$$

Equation (1) is simplified to be:

$$m\ddot{x}_2 + C_5\dot{X}_2 + K_5X_2 - K_5T\theta - C_5T\dot{\theta} = P_C \tag{2}$$

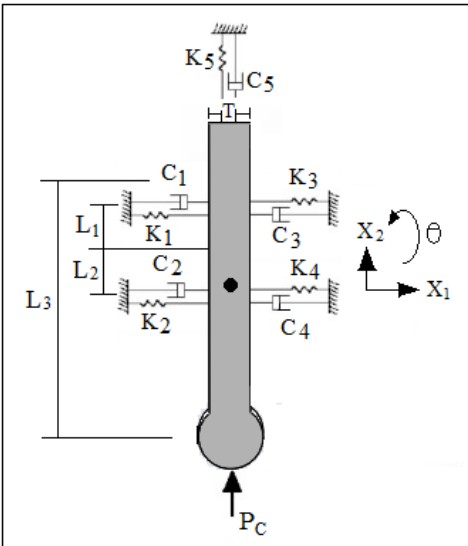

**Figure 1.** Vibration of the follower stem.

The Newton equation of rigid body dynamics when the follower translates in the horizontal direction is:

$$\sum \text{Forces} = m\ddot{x}_1$$

$$-K_1(X_1 - L_1\theta) - C_1\left(\dot{X}_1 - L_1\dot{\theta}\right) - K_3(X_1 - L_1\theta) - C_3\left(\dot{X}_1 - L_1\dot{\theta}\right) - K_2(X_1 - L_2\theta) - C_2\left(\dot{X}_1 - L_2\dot{\theta}\right) - K_4(X_1 - L_2\theta) - C_4\left(\dot{X}_1 - L_2\dot{\theta}\right) = m\ddot{x}_1 \tag{3}$$

Equation (3) is simplified to be:

$$m\ddot{x}_1 + (C_1 + C_2 + C_3 + C_4)\dot{X}_1 - (C_1L_1 + C_2L_2 + C_3L_1 + C_4L_2)\dot{\theta} + (K_1 + K_2 + K_3 + K_4)X_1 - (K_1L_1 + K_2L_2 + K_3L_1 + K_4L_2)\theta = 0 \tag{4}$$

The Euler equation of rigid body dynamics when the follower rotates about z-direction is:

$$\sum \text{Moments} = I\ddot{\theta}$$

$$P_C\frac{L_3}{2}\text{Sin}(\theta) - mg\frac{L_3}{2}\text{Sin}(\theta) - K_1(L_1\theta - X_1)L_1 - C_1\left(L_1\dot{\theta} - \dot{X}_1\right)L_1 - K_3(L_1\theta - X_1)L_1 - C_3\left(L_1\dot{\theta} - \dot{X}_1\right)L_1 - L_2(L_2\theta - X_1)L_2 - C_2\left(L_2\dot{\theta} - \dot{X}_1\right)L_2 - L_4(L_2\theta - X_1)L_2 - C_4\left(L_2\dot{\theta} - \dot{X}_1\right)L_2 = I\ddot{\theta} \tag{5}$$

Equation (5) is simplified to be

$$\ddot{\theta} + (C_1 + C_3)L_1^2\dot{\theta} + (C_2 + C_4)L_2^2\dot{\theta} + (K_1 + K_3)L_1^2\theta + (K_2 + K_4)L_2^2\theta - P_C\frac{L_3}{2}\theta + mg\frac{L_3}{2}\theta - (C_1 + C_3)L_1\dot{X}_1 - (C_2 + C_4)L_2\dot{X}_1 - (K_1 + K_3)L_1X_1 - (K_2 + K_4)L_2X_1 = 0 \tag{6}$$

where:

$$I = \frac{mL_3^2}{12} + m\frac{L_3^2}{4}$$

Put the equations of motion Equations (2), (4), and (6) in matrix form:

$$\begin{bmatrix} m_{11} & m_{12} & m_{13} \\ m_{21} & m_{22} & m_{23} \\ m_{31} & m_{32} & m_{33} \end{bmatrix} \begin{Bmatrix} \ddot{X}_1 \\ \ddot{X}_2 \\ \ddot{\theta} \end{Bmatrix} + \begin{bmatrix} C_{11} & C_{12} & C_{13} \\ C_{21} & C_{22} & C_{23} \\ C_{31} & C_{32} & C_{33} \end{bmatrix} \begin{Bmatrix} \dot{X}_1 \\ \dot{X}_2 \\ \dot{\theta} \end{Bmatrix} + \begin{bmatrix} K_{11} & K_{12} & K_{13} \\ K_{21} & K_{22} & K_{23} \\ K_{31} & K_{32} & K_{33} \end{bmatrix} \begin{Bmatrix} X_1 \\ X_2 \\ \theta \end{Bmatrix} = \begin{Bmatrix} F_1 \\ F_2 \\ F_3 \end{Bmatrix} \tag{7}$$

where:

$$m_{11} = m, \ m_{12} = 0, \ m_{13} = 0$$

$$m_{21} = 0, \ m_{22} = m, \ m_{23} = 0$$

$$m_{31} = 0, \ m_{32} = 0, \ m_{33} = I$$

$$C_{11} = C_1 + C_2 + C_3 + C_4, \ C_{12} = 0, \ C_{13} = -(C_1L_1 + C_2L_2 + C_3L_1 + C_4L_2)$$

$$C_{21} = 0, \ C_{22} = C_5, \ C_{23} = -C_5T$$

$$C_{31} = -(C_1 + C_3)L_1 - (C_2 + C_4)L_2, \ C_{32} = 0, \ C_{33} = (C_1 + C_3)L_1^2 + (C_2 + C_4)L_2^2$$

$$K_{11} = (K_1 + K_2 + K_3 + K_4), \ K_{12} = 0, \ K_{13} = -(K_1L_1 + K_2L_2 + K_3L_1 + K_4L_2)$$

$$K_{21} = 0, \ K_{22} = K_5, \ K_{23} = -K_5T$$

$$K_{31} = -(K_1 + K_3)L_1 - (K_2 + K_4)L_2, \ K_{32} = 0, \ K_{33} = (K_1 + K_3)L_1^2 + (K_2 + K_4)L_2^2 - P_C\frac{L_3}{2} + mg\frac{L_3}{2}$$

$$F_1 = 0, \ F_2 = P_C, \ F_3 = 0$$

By substituting Equation (7) in terms of an impedance matrix as in below [11]:

$$\begin{bmatrix} Z_{11}(\Omega) & 0 & Z_{13}(\Omega) \\ 0 & Z_{22}(\Omega) & Z_{23}(\Omega) \\ Z_{31}(\Omega) & 0 & Z_{33}(\Omega) \end{bmatrix} \begin{Bmatrix} X_1 \\ X_2 \\ \theta \end{Bmatrix} = \begin{Bmatrix} F_1 \\ F_2 \\ F_3 \end{Bmatrix} \tag{8}$$

where:

$$Z_{11}(\Omega) = -\Omega^2 m_{11} + i\Omega C_{11} + K_{11}$$

$$Z_{13}(\Omega) = i\Omega C_{13} + K_{13}$$

$$Z_{22} = -\Omega^2 m_{22} + i\Omega C_{22} + K_{22}$$

$$Z_{31} = i\Omega C_{31} + K_{31}$$

$$Z_{33} = -\Omega^2 m_{33} + i\Omega C_{33} + K_{33}$$

By solving the Eigenvalue problem as in below:

$$\begin{vmatrix} Z_{11}(\Omega) & 0 & Z_{13}(\Omega) \\ 0 & Z_{22}(\Omega) & Z_{23}(\Omega) \\ Z_{31}(\Omega) & 0 & Z_{33}(\Omega) \end{vmatrix} = 0 \tag{9}$$

By solving the Eigenvectors problem to obtain the amplitude of vibration as in below:

$$\begin{Bmatrix} X_1 \\ X_2 \\ \theta \end{Bmatrix} = \begin{bmatrix} Z_{11}(\Omega) & 0 & Z_{13}(\Omega) \\ 0 & Z_{22}(\Omega) & Z_{23}(\Omega) \\ Z_{31}(\Omega) & 0 & Z_{33}(\Omega) \end{bmatrix}^{-1} \begin{Bmatrix} F_1 \\ F_2 \\ F_3 \end{Bmatrix} \tag{10}$$

By solving Equation (10) to obtain the nonlinear response of the follower as in below:

$$\begin{Bmatrix} x_1(t) \\ x_2(t) \\ \theta(t) \end{Bmatrix} = \begin{Bmatrix} X_1 \\ X_2 \\ \theta \end{Bmatrix} Sin(\Omega t - \emptyset) \tag{11}$$

The contact force at the contact point between the cam and the follower is illustrated as in below [12]:

$$P_C = \frac{1}{Cos(\emptyset_1)} \left[ K(\Delta + X(t)) - KX(t) - m\ddot{X}(t) \right] \tag{12}$$

The pressure angle is as in below [13,14]:

$$\tan(\emptyset_1) = \frac{\dot{X}(t)}{X(t) + R_b^2}$$

The nonlinear response of the follower in the horizontal and vertical directions is:

$$X(t) = \sqrt{X_1^2(t) + X_2^2(t)} \tag{13}$$

The contact stress is calculated using the theory of contact mechanics as in below [15]:

$$\sigma_C = -0.591 \sqrt{\frac{P_C \left( \frac{1}{R_1} + \frac{1}{R_2} \right)}{L\Delta}} \tag{14}$$

## 3. Numerical Simulation

The SolidWorks program was used to find the kinetic parameters of the dynamic motion such as displacement, velocity, and acceleration for the follower [16]. The follower was moved with three degrees of freedom while the multi degrees of freedom were moved

up and down, as indicated in Figure 2. The solution of the nonlinear response of the follower was divided into 1000 frames per second to make the dynamic motion of the follower smoother. The cam profile was formed from the nose, flank no.(1), base circle, and flank no.(2).

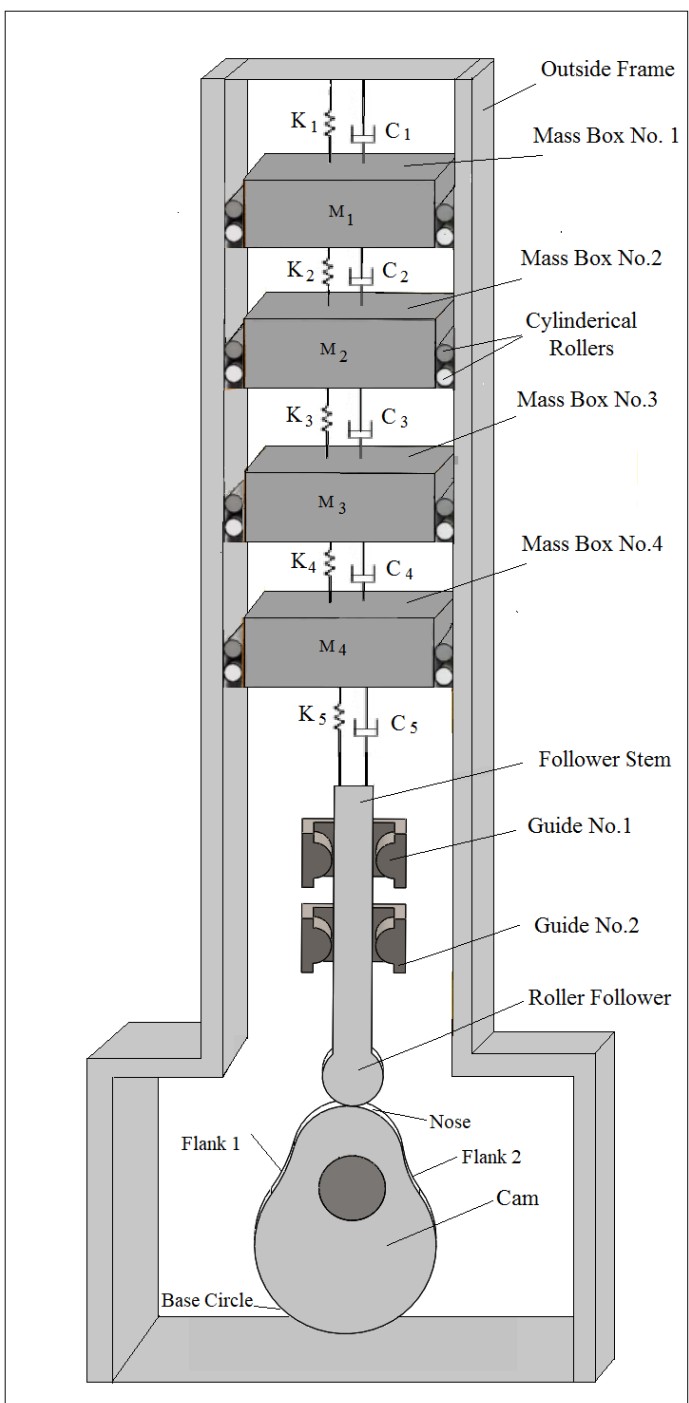

**Figure 2.** Cam follower system with multi degrees of freedom.

Figure 3 shows the three involutes' profiles of the cam with its dimensions. The follower nonlinear response, velocity, and acceleration are shown in Figures 4–6 for the three selected profiles. The system with cam speed (N = 200 rpm) and internal distance of the follower guide from inside (I.D. = 16 mm) were considered in the dynamic kinematic relationship calculations.

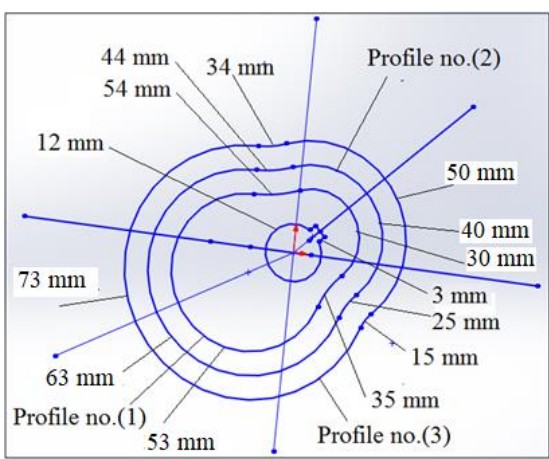

**Figure 3.** Three involutes' profile of the cam with its dimension.

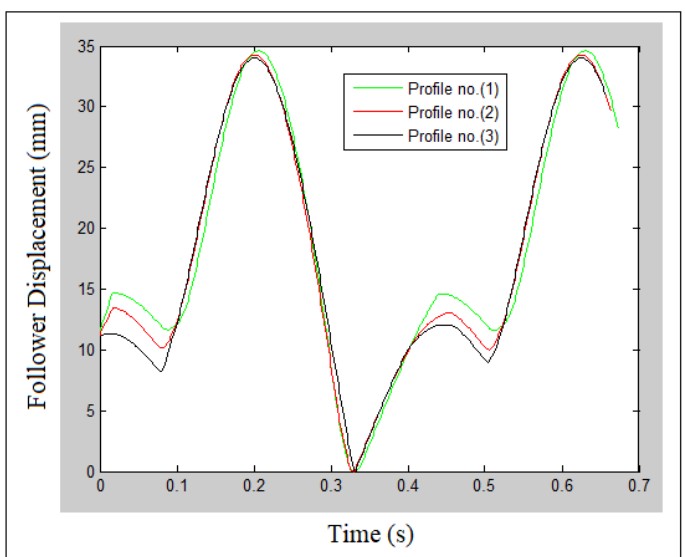

**Figure 4.** Nonlinear response against time for the three involutes' profiles of the cam.

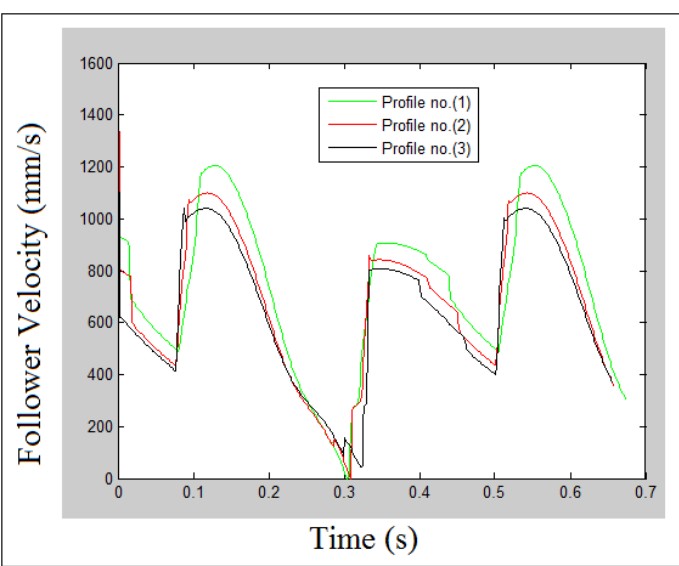

**Figure 5.** Velocity against time for the three involutes' profiles of the cam.

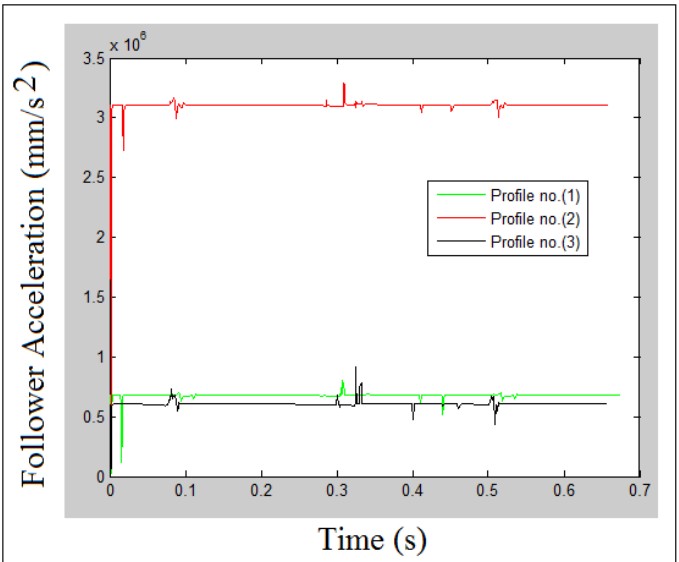

**Figure 6.** Acceleration against time for the three involutes' profiles of the cam.

## 4. Finite Element Analysis Steps Using SolidWorks Program

Mesh generation is a very important step in the calculation of the bending deflection and the contact stress distributions at the contact point and around the square grooving key [17]. The SolidWorks program was used in the mesh generation based on the global element size, tolerance, and local mesh control specifications. A parabolic tetrahedral element was used to mesh the cam-follower mechanism system, which had four corner nodes, six mid-side nodes, and six edges. Figure 7 shows the mesh generation of the cam-follower mechanism.

Always, the convergence test is needed to determine the size of the elements at which the value of deflection and stress settle down. Finite element analysis of the convergence curve defines the relationship between any quantities against the total number of elements. Figures 8 and 9 show the convergence test of the bending deflection and the contact stress for cam profile no.(1). The system with the internal dimension of the follower guide from inside (I.D. = 16 mm) and cam speed (N = 200 rpm) was considered in the convergence test. It can be concluded from Figures 8 and 9 that the quantity of the bending deflection and the contact stress has increased with the increase in the total number of elements while the element size has decreased. Tetrahedral elements with edge sizes of 3.5 mm, 3 mm, 2.5 mm, 2 mm, 1.5 mm, and 1 mm were selected in the convergence test. Von Mises criterion was used in the calculation of maximum contact stress distribution since Von Mises represents the equivalent of the stresses in the x, y, and z directions.

The steps that describe finite element analysis using the SolidWorks program are listed below:

(a) Import the (pearcam.SLDPRT and squaregroovingkey.SLDPRT) parts for the cam and square grooving key geometries, respectively.

(b) Select the Mate button to make the mates between the three sides of the pear cam and the three sides of the square grooving key.

(c) Select the Connections button and choose Contact Sets to create contact between the three surfaces of the pear cam from inside and the three surfaces of the square grooving key from outside.

(d) Select the Fixtures button and select Fixed Geometry to set the clamped boundary conditions at the hub radius (r = 12 mm) and the last surface of the square grooving key.

(e) Select the External Loads button and choose Pressure since the value of the contact pressure is varied based on the contact with the nose, base circle, flank no.(1), and flank no.(2).

(f)　Select the Mesh button and choose Create Mesh to carry out the mesh generation between the cam and square grooving key. Choose Details and Total Elements to describe the total number of elements. Keep changing the element size and the total number of elements to accomplish the convergence test of the bending deflection and the contact stress quantities.

(g)　Select the Simulation button to run the finite element model.

(h)　Select the Results button to see the contour of the bending deflection and the contact stress distributions.

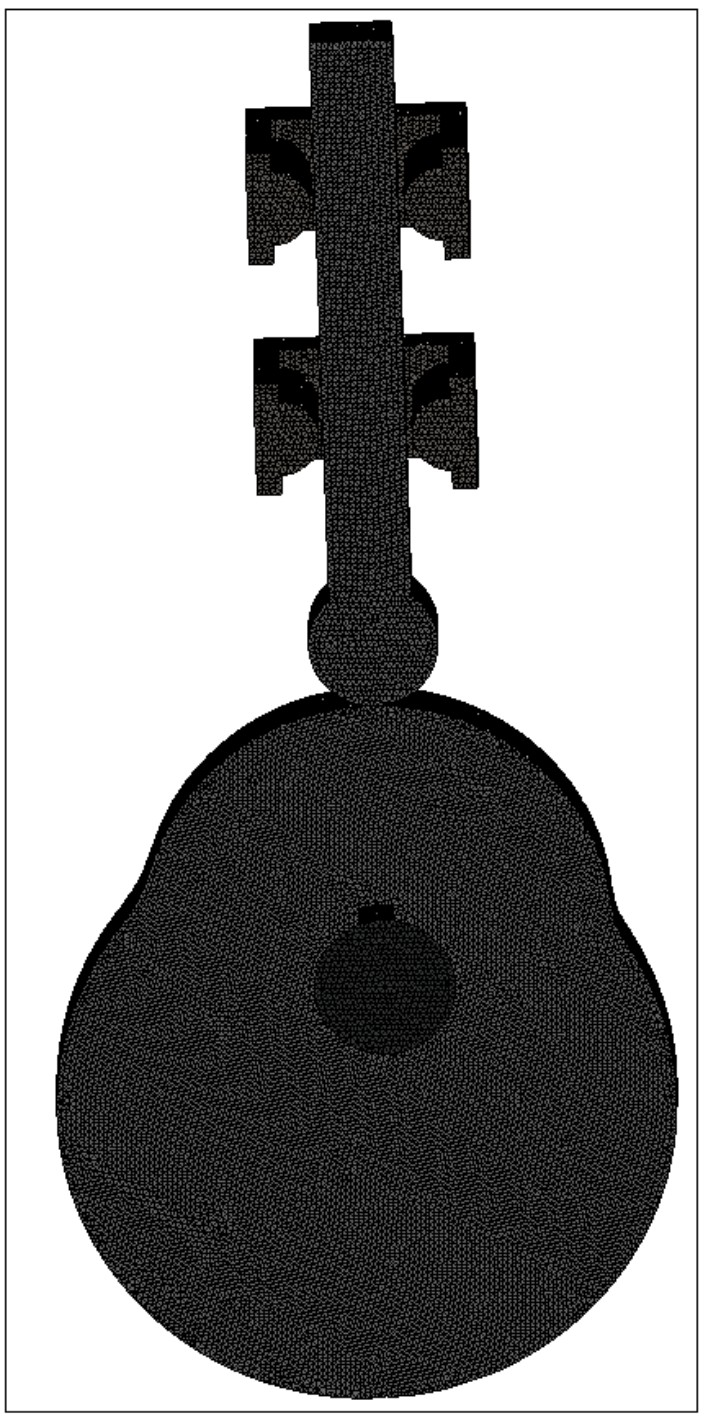

**Figure 7.** Mesh generation of cam-follower mechanism for profile no.(3).

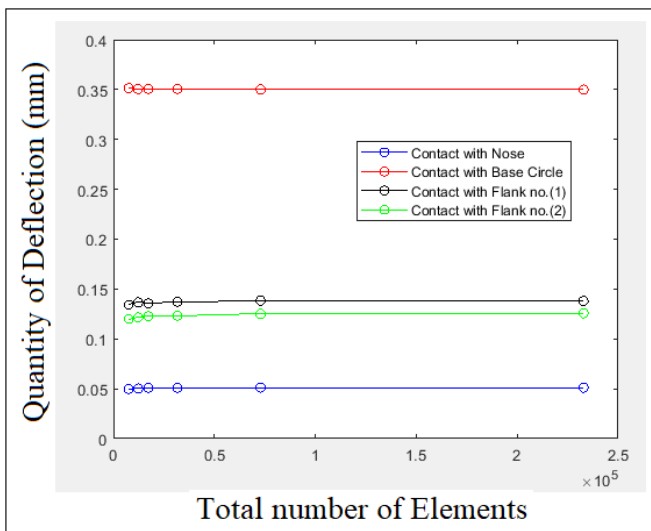

**Figure 8.** Convergence test of the bending deflection quantity at different contact locations for profile no.(1).

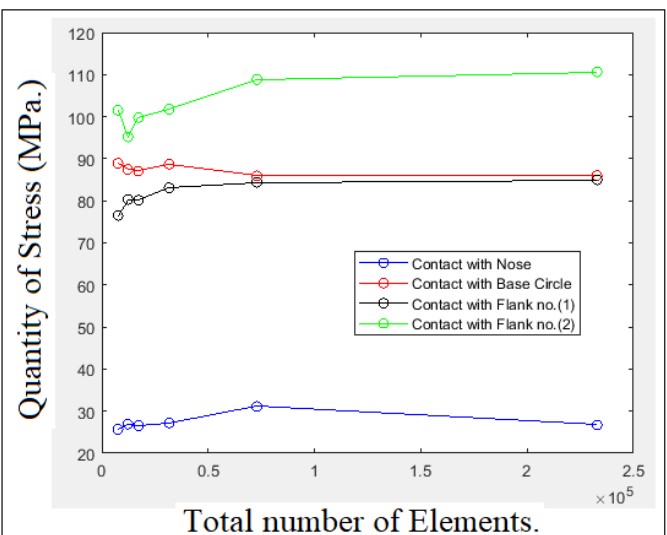

**Figure 9.** Convergence test of the contact stress quantity at different contact locations for profile no.(1).

There are three options for contact using the SolidWorks program as described below:

(1)    Bonded: The two parts act as one part and the two parts behave as if they are welded.
(2)    No Penetration: This type of contact prevents interface between the two parts, but allows gaps to form. This is the most time-consuming option to solve.
(3)    Allow Penetration: This option treats the two parts as disjointed and the loads are allowed to cause interference between parts.

In other finite element programs such as the ANSYS software, the only option available is bonded. No sliding or separation between faces or edges is allowed. In this article, the SolidWorks program is more suitable than other programs of finite element analysis since the separation and sliding between the cam and the follower are allowed at high speeds.

## 5. Methods of Experiment Setup

The pear cam was installed on the aluminum shaft through the square grooving key in which the total assembly was installed on the bottom ruler while the roller follower was pinned at the upper ruler as shown in Figure 10.

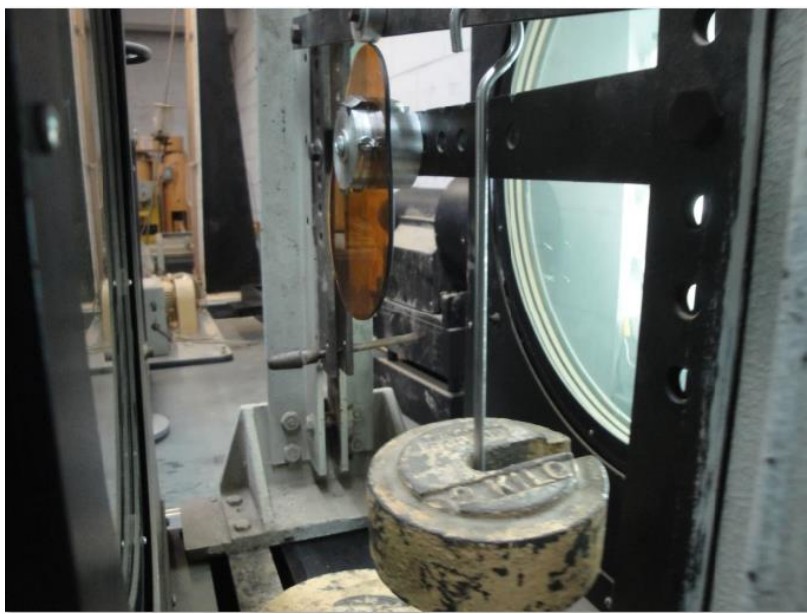

**Figure 10.** Cam installation on the shaft through square grooving key.

ASTM (D790) for three bending flexural tests was used to determine the stress fringe constant. A specimen of (PMMA) material with the dimensions (length = 80 mm, Wide = 10 mm, and thickness = 4 mm) was prepared for the three bending point test. The three bending point test was placed inside the photo-elastic device since the plane polariscope with a dark field was considered to show the stress distribution. Figure 11 shows the fringe intensification of the three bending point tests using the photo-elastic device.

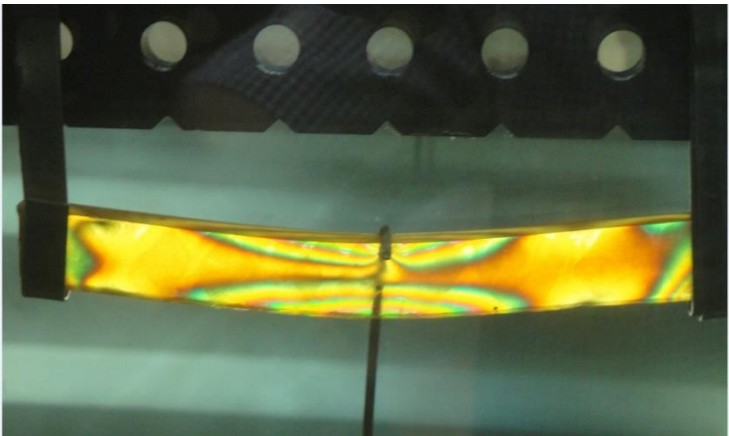

**Figure 11.** Three bending point test.

Due to the contact between the cam and the follower, there will be two components of the contact force based on the pressure angle. After applying the maximum shear stress theory of the Tresca criterion, the principal stresses in the horizontal and vertical directions can be calculated, and due to the difference between these two stresses, the fringe order appeared in the photo-elastic device. The difference between the principal stresses represents the experiment value of the contact stress, which is just one numeric value.

$$f_\sigma = \frac{h(\sigma_1 - \sigma_2)}{N} \qquad (15)$$

where:

$$\sigma_1 - \sigma_2 = \frac{My}{I_1} = \frac{F_s L_o y}{4I_1}$$

The numbers (1), (4), (7), and (10) represent the contact points between the cam and the follower at four contact locations of the cam with the follower. The numbers (2), (3), (5), (6), (8), (9), (11), and (12) represent the contact points between the cam and square grooving key. The contact stress was calculated on these mentioned points above. Figures 12–15 show the contact between the cam and the follower and between the cam and the square grooving key at the nose, flank no.(1), base circle, and flank no.(2), respectively.

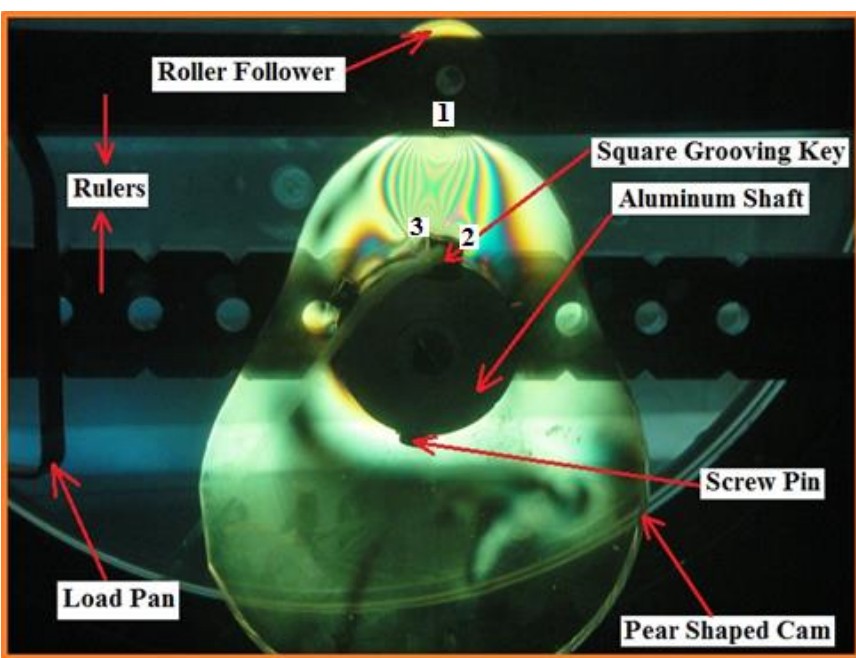

**Figure 12.** Experimental analysis of the stress distribution when the follower comes in contact with the nose.

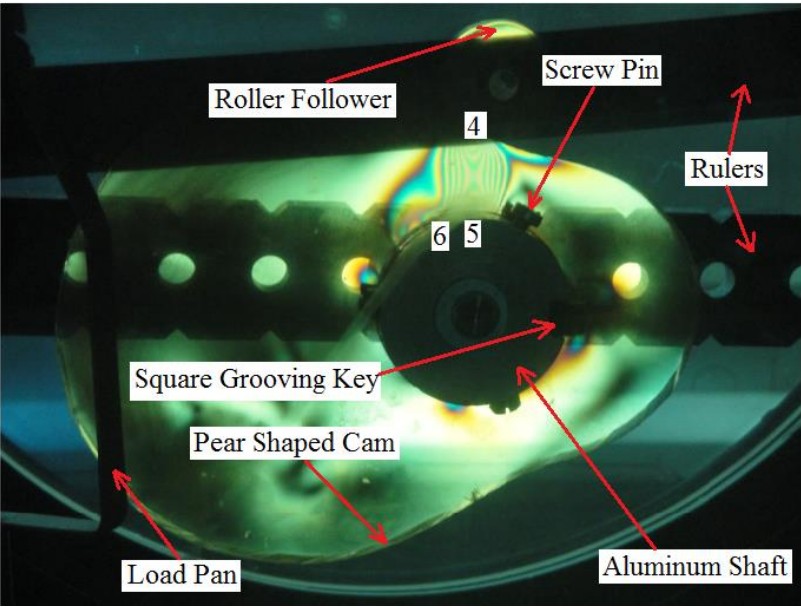

**Figure 13.** Experimental analysis of the stress distribution when the follower comes in contact with flank no.(1).

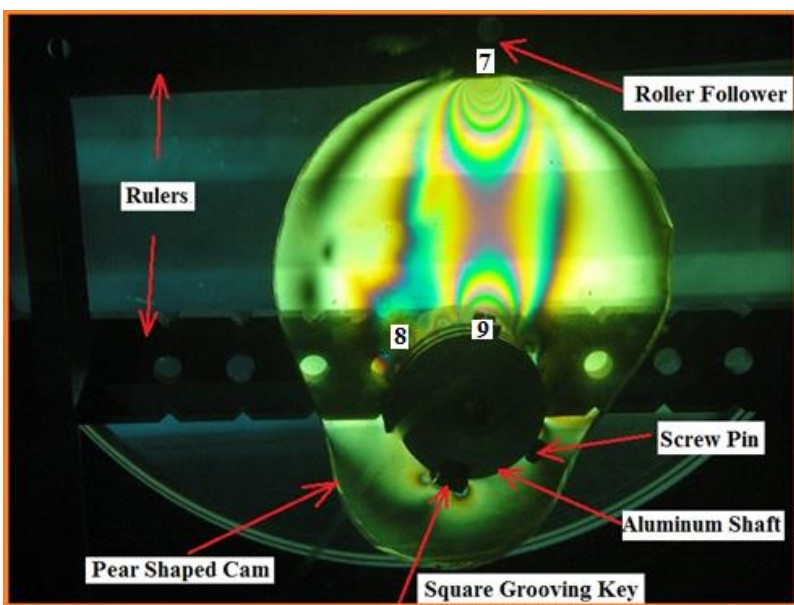

**Figure 14.** Experimental analysis of the stress distribution when the follower comes in contact with the base circle.

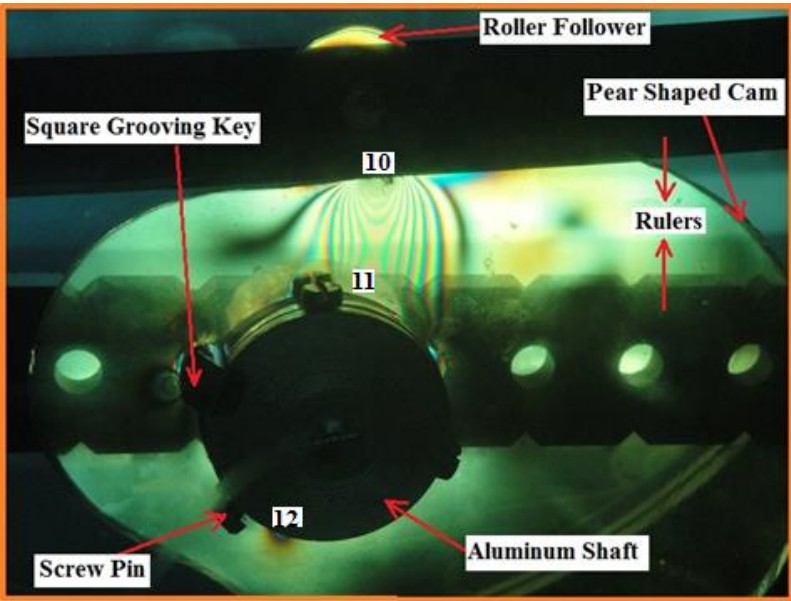

**Figure 15.** Experimental analysis of the stress distribution when the follower comes in contact with the flank no.(2).

The internal distance of the follower guide from the inside was assumed to be (I.D. = 16 mm) since it is constant from the manufacturer. The experiment apparatus with the code OPTOTRAK 30/20 had a high-speed camera in the foreground. There were two vertical grooves inside the main frame box in which the cylindrical rollers move up and down to absorb the potential energy of the follower. Multi degrees of freedom (spring-damper-mass) systems were used on the follower stem to reduce the detachment between the cam and the follower and to reduce the maximum contact stress. The follower was moved with three degrees of freedom (up-down, right-left, and rotation about the *z*-axis). Figure 16 shows the experiment test.

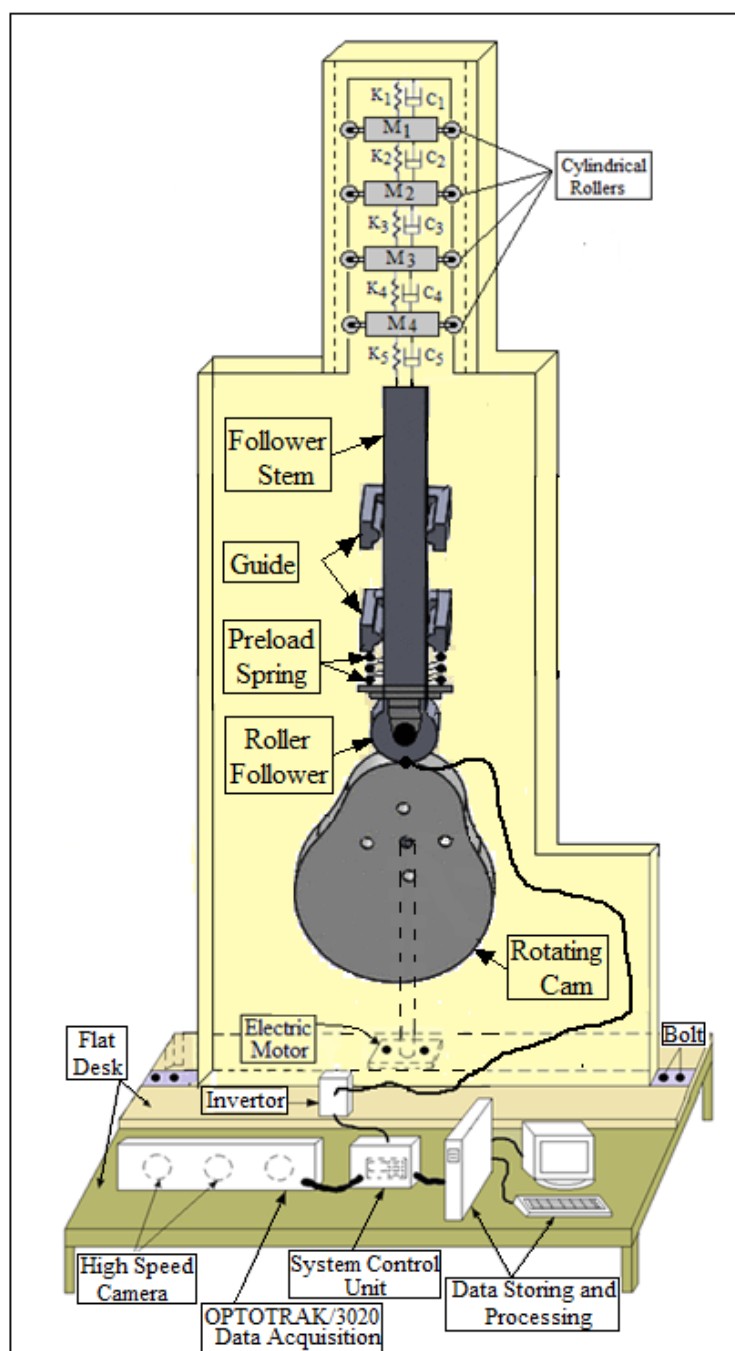

**Figure 16.** Experiment setup of cam-follower mechanism.

## 6. Largest Lyapunov Exponent Parameter

In the design, the largest Lyapunov exponent parameter was one of the indicators to detect separation and chaos between the cam and the follower through the contact stress. The more the contact stress between the cam and the follower, the more the value of the largest Lyapunov exponent. When the value of the Lyapunov exponent is positive, it means that the value of contact stress is very high (non-periodic motion). The negative largest Lyapunov exponent value indicates periodic motion (the cam and the follower are in permanent contact) and the contact stress has a minimum value. The Wolf algorithm code, based on MATLAB software, was used to extract the values of the largest Lyapunov exponent by

monitoring the orbital divergence of the follower displacement [18]. Equations (16) and (17) were used to build the Wolf algorithm code of the dynamic tool [19].

$$d(t) = De^{\lambda t} \tag{16}$$

$$y(i) = \frac{1}{\Delta t}\left[\ln d_j(i)\right] \tag{17}$$

The average logarithmic divergence approach was used to extract the value of the largest Lyapunov exponent of the contact stress. The straight line represents the slope of the logarithmic function of the contact stress, which gives the value of the Lyapunov exponent parameter using the Least Square Method of Curve Fitting. The nonlinear curve represents the logarithmic function of the contact stress against time. The analytic solution of the largest Lyapunov exponent was found after using Equation (13) in the Wolf algorithm code. The experiment value of the Lyapunov exponent was calculated after tracking the follower position using a high-speed camera in the foreground of the OPTOTRAK 30/20 device while the numerical simulation of the Lyapunov exponent was determined after using the nonlinear response of the follower from the SolidWorks program in the Wolf algorithm code. Figure 17 shows the comparison of the average logarithmic divergence of the contact stress against time for (I.D. = 16 mm) and (N = 500 rpm) for cam profile no.(3).

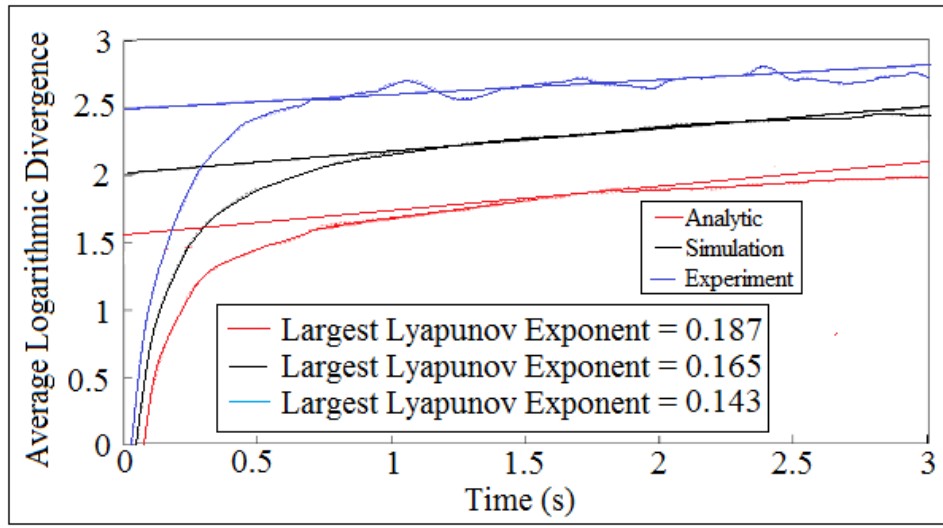

**Figure 17.** Average logarithmic divergence against time for (I.D. = 16 mm) and (N = 500 rpm) for profile no.(3).

## 7. Phase-Plane Diagram

The phase-plane diagram shows how the fringe order of the contact stress distribution at the contact point grows or shrinks over time. When the orbit of the fringe order is one closed cycle, it means that the motion of the contact stress is periodic. When the attractor of the fringe order diverges with no limit of spiral cycles, it indicates that the motion of the contact stress is non-periodic. Figure 18 shows the phase-plane mapping of the fringe order of the contact stress at different cam speeds and different internal distances of the follower guide from inside for cam profile no.(1). It can be concluded from the phase-plane diagram that the motion of the contact stress is periodic since the attractor of the fringe order is one closed cycle as indicated in Figure 18d,g. The other systems indicate non-periodic motion and chaos since the attractor of the fringe order of the contact stress diverges with no limit of spiral cycles.

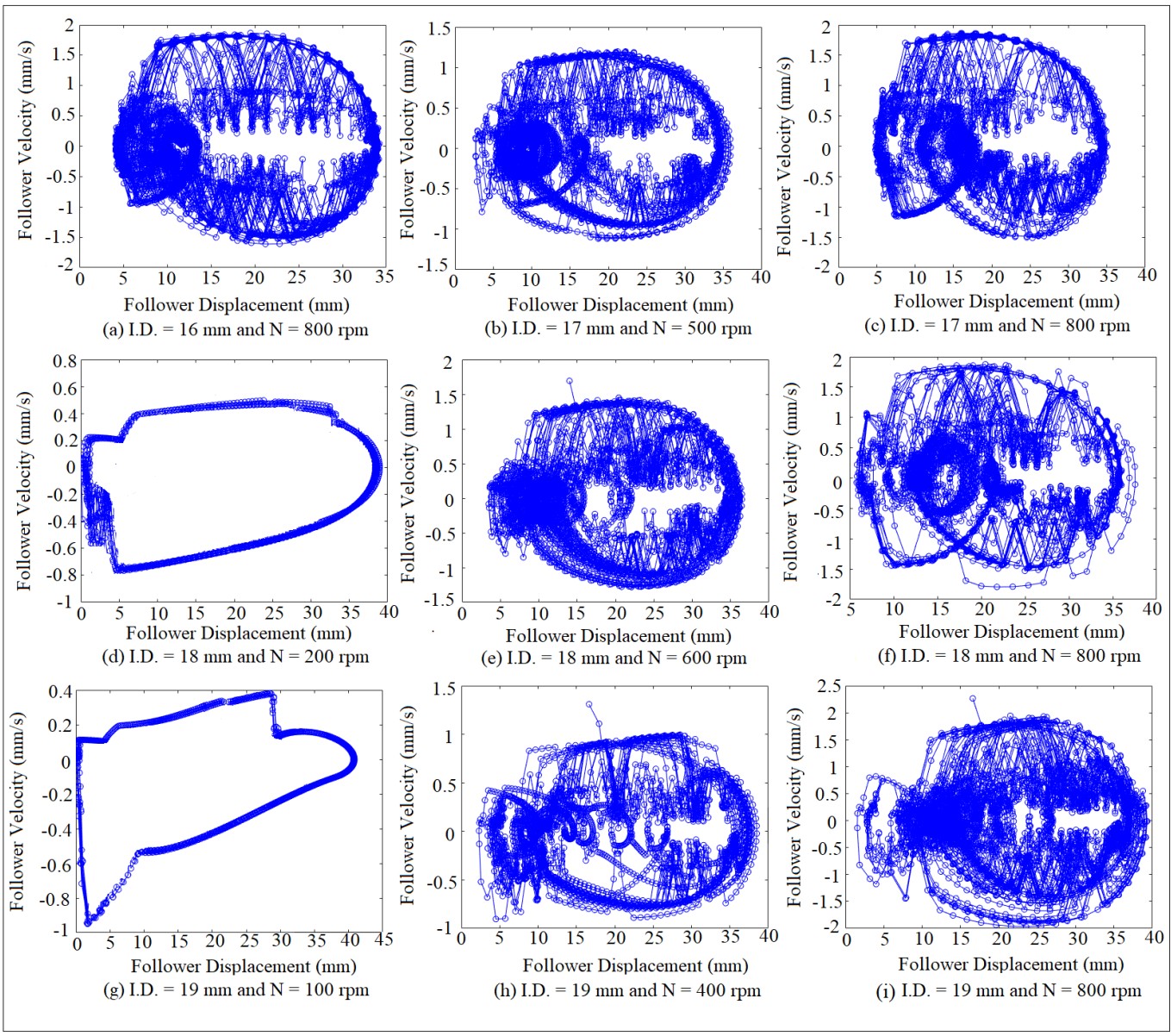

**Figure 18.** Phase-plane mapping for cam profile no.(1) at different (I.D.) and different (N).

## 8. Results and Discussions

Figure 19 shows the comparison of the nonlinear response of the follower at (I.D. = 16 mm) and (N = 200 rpm) for cam profile no.(1). The analytic nonlinear response of the follower was calculated after applying Equation (13) while the numerical simulation of the nonlinear response of the follower was determined using the SolidWorks program. The experiment value of the nonlinear response of the follower was obtained after catching the follower position using the OPTOTRAK 30/20 device through a high-speed camera. The dwell stroke is tangible for the analytic solution and for the experiment setup while the dwell stroke disappeared for the numerical simulation of the nonlinear response of the follower.

Figures 20–22 show the bending deflection contour for cam profile no.(1), no.(2), and no.(3), respectively, at different contact locations of the follower with the cam. The value of the bending deflection is very high when the follower comes in contact with the base circle for cam profile no.(1) and no.(3) since there is no effect when the cam comes in contact with the square grooving key as illustrated in Figures 20b and 22c, respectively. The value of bending deflection is very low when the follower comes in contact with the nose and base circle for cam profile no.(2). The value of the bending deflection is very high at the nose

location for cam profile no.(3). In the design, cam profile no.(2) can be selected because the bending deflection value is at the lowest level.

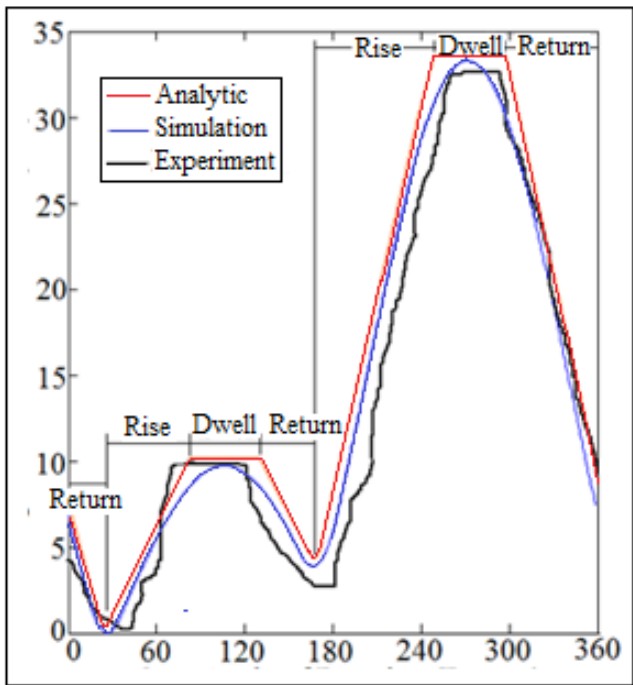

**Figure 19.** Comparison of nonlinear response of the follower.

Figures 23–25 show the Von Mises stress contours for cam profile no.(1), no.(2), and no.(3) at different contact locations for the cam with the follower. Four contact locations were considered such as (0 degrees at the nose, 90 degrees at flank no.(1), 180 degrees at the base circle, and 270 degrees at flank no.(2). Von Mises stress was used in the finite element analysis since Von Mises represents the equivalent of the stresses in x, y, and z directions. There is a relationship between contact stress and bending deflection. The more the value of contact stress, the more the value of bending deflection. The value of contact stress for cam profile no.(2) is smaller than the others since the bending deflection at the nose and at the base circle was decreased. Around the square grooving key, the contact stress has a maximum value for cam profile no.(3), while the contact stress has a minimum value for cam profile no.(2) at the nose location. Flank no.(1) and flank no.(2) have a concave negative radius of curvatures. The value of the contact stress at flank no.(2) is higher than the value of the contact stress at flank no.(1) because of the bigness of the radius of curvature of flank no.(2) for the three involutes of the cam profile. The contact stress for flank no.(1) and flank no.(2) have minimum values if it is compared with the contact stress values at the nose and at the base circle locations. The data of the contact stress against time around these contact locations were selected to study the relationship between the contact stress and Lyapunov exponent parameter. The more the value of contact stress, the more the value of the Lyapunov exponent parameter. It can be concluded from Figure 25 that the follower motion at the nose location has non-periodic motion and chaos, while at the flank no.(1) location, the follower motion could be either periodic or quasi-periodic for cam profile no.(3). The follower motion is non-periodic and chaotic when the follower comes in contact with flank no.(2) while the follower motion is periodic since the contact stress is at the lowest level for cam profile no.(1) as shown in Figure 23. The non-periodic motion of the follower is at the lowest level for all the contact locations for cam profile no.(2) as depicted in Figure 24.

Table 1 shows the comparison of the contact stress at different contact locations of the follower with cam profile no.(1). Points (1), (4), (7), and (10) represent the contact points between the cam and the follower while points (2), (3), (5), (6), (8), (9), (11), and (12) reflect

the contact between the cam and the square grooving key. The value of the contact stress at the point of contact between the cam and the follower is higher than the value of the contact stress around the square grooving key because points (1), (4), (7), and (10) are very close to the applied load. The contact stress when the follower comes in contact with the base circle for cam profile no.(1) is higher than the others because of the bigness of the radius of curvature of the base circle. The numerical simulation of the contact stress was done using the SolidWorks program while the experiment value of the contact stress was carried out using the photo-elastic device.

**Table 1.** Comparison of contact stress at different contact locations for cam profile no.(1).

| Contact Points | | Experiment (MPa) | Numerical (MPa) | Error (%) |
|---|---|---|---|---|
| | 1 | 23.944 | 26.708 | 10.348 |
| Contact with the Nose | 2 | 8.089 | 8.903 | 9.143 |
| | 3 | 14.079 | 15.58 | 9.634 |
| | 4 | 78.44 | 84.972 | 7.687 |
| Contact with Flank no.(1) | 5 | 16.145 | 17.702 | 8.795 |
| | 6 | 12.445 | 14.162 | 12.123 |
| | 7 | 76.88 | 85.999 | 10.603 |
| Contact with the Base Circle | 8 | 26.5 | 28.66 | 7.536 |
| | 9 | 19.33 | 21.502 | 10.101 |
| | 10 | 92.243 | 110.652 | 16.636 |
| Contact with Flank no.(2) | 11 | 16.07 | 18.46 | 12.946 |
| | 12 | 19.242 | 21.614 | 10.974 |

Figure 26 shows the variation of the contact stress against the angle of contact at different involutes' profiles for the cam. The contact stress is decreased with the increasing angle of contact for cam profile no.(2) and cam profile no.(3). The contact stress is varied sinusoidally against the angle of contact for cam profile no.(1). It can be concluded from Figure 26 that the nonlinear dynamics phenomenon of the follower is at the highest level on the nose location for cam profile no.(3), while the lowest level of the nonlinear dynamics phenomenon of the follower occurred at the base circle location for cam profile no.(1) and cam profile no.(2).

Figures 27–29 show the largest Lyapunov exponent parameter against cam speeds at different internal distances of the follower guide from inside for cam profile no.(1), no.(2), and no.(3). The largest Lyapunov exponent is varied sinusoidally with the increasing of cam speeds (N) for all internal distance of the follower guide from inside (I.D.). The largest Lyapunov exponent settles down at cam speeds (N = 500–700 rpm) for (I.D. = 16, 17, and 18 mm) as indicated in Figure 27. The largest Lyapunov exponent is varied sinusoidally against cam speeds at (N = 100–400 rpm) for all systems. The largest Lyapunov exponent settles down at cam speeds (N = 600 rpm and 700 rpm) for (I.D. = 16, 18, and 19 mm) as indicated in Figure 28. Moreover, the largest Lyapunov exponent is varied sinusoidally with the increasing of cam speeds for (I.D. = 16, 18, and 19 mm), while the system with (I.D. = 17 mm), the largest Lyapunov exponent is decreased with the increasing of cam speeds until it settles down at cam speeds (N = 600–800 rpm) as indicated in Figure 29. The largest peak of the largest Lyapunov exponent is (LLE = 8.5) at (I.D. = 16 mm) and (N = 300 rpm) for cam profile no.(1) while the smallest peak of the largest Lyapunov exponent is (LLE = 0.23) at (I.D. = 16 mm) and (N = 500 rpm) for cam profile no.(3). Cam profile no.(2) has smaller values of the largest Lyapunov exponent than the other systems in which it can be selected in the design.

Figures 30 and 31 show the phase-plane mapping of the fringe order of the contact stress at different cam speeds (N) and different internal distances of the follower guide

from inside (I.D.) for cam profile no.(2) and cam profile no.(3). It can be concluded from the phase-plane diagram that the motion of the contact stress is periodic since the attractor of the fringe order is one closed cycle as indicated in Figures 30a and 31a. The other systems indicate non-periodic motion and chaos since the attractor of the fringe order of the contact stress diverges with no limit of spiral cycles. In general, the cross-linking of the fringe order of the contact stress is increased with the increasing of cam speeds (N) and internal distance of the follower guide from the inside (I.D.).

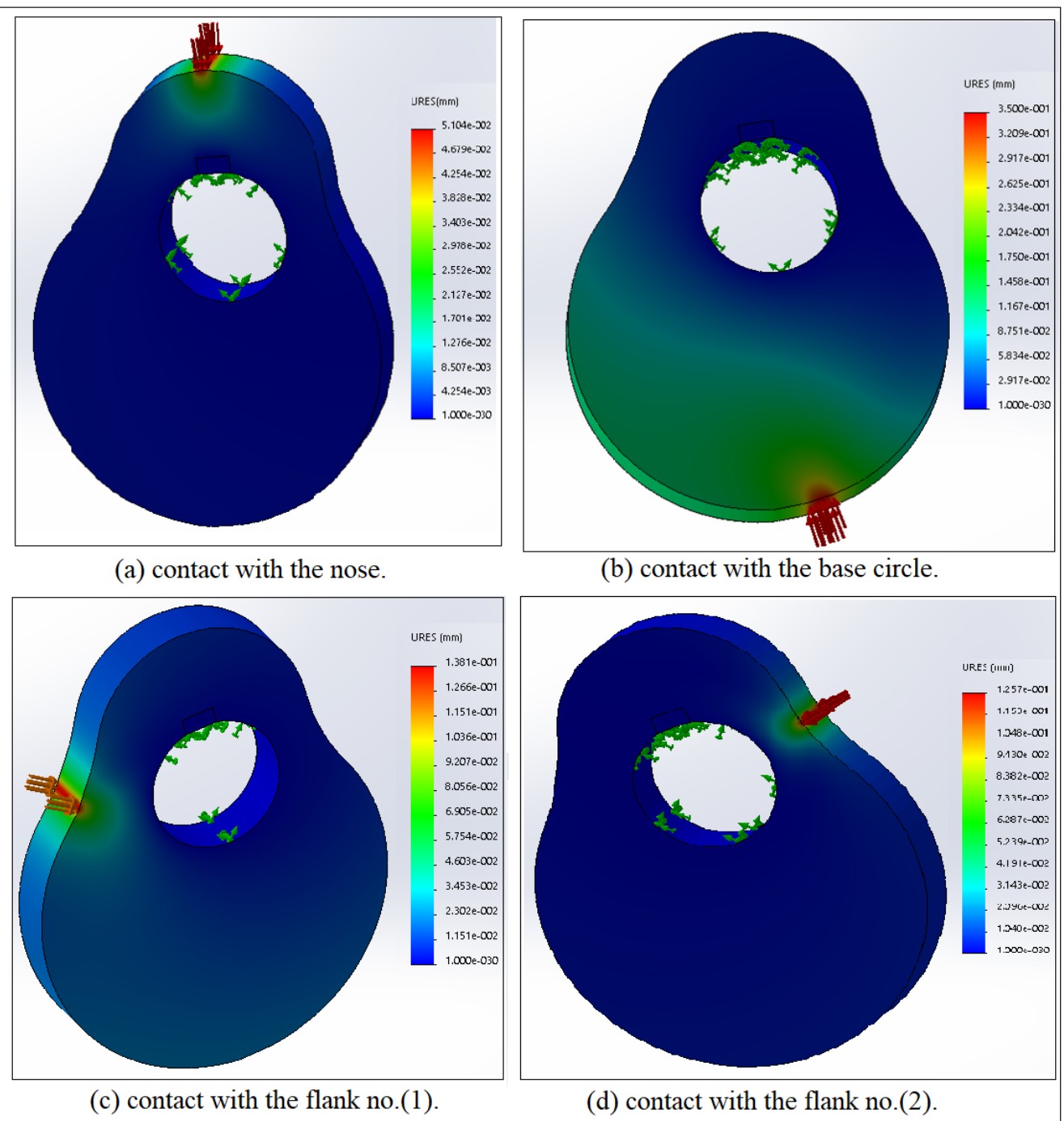

**Figure 20.** Bending deflection mapping contour for profile no.(1) at different contact locations.

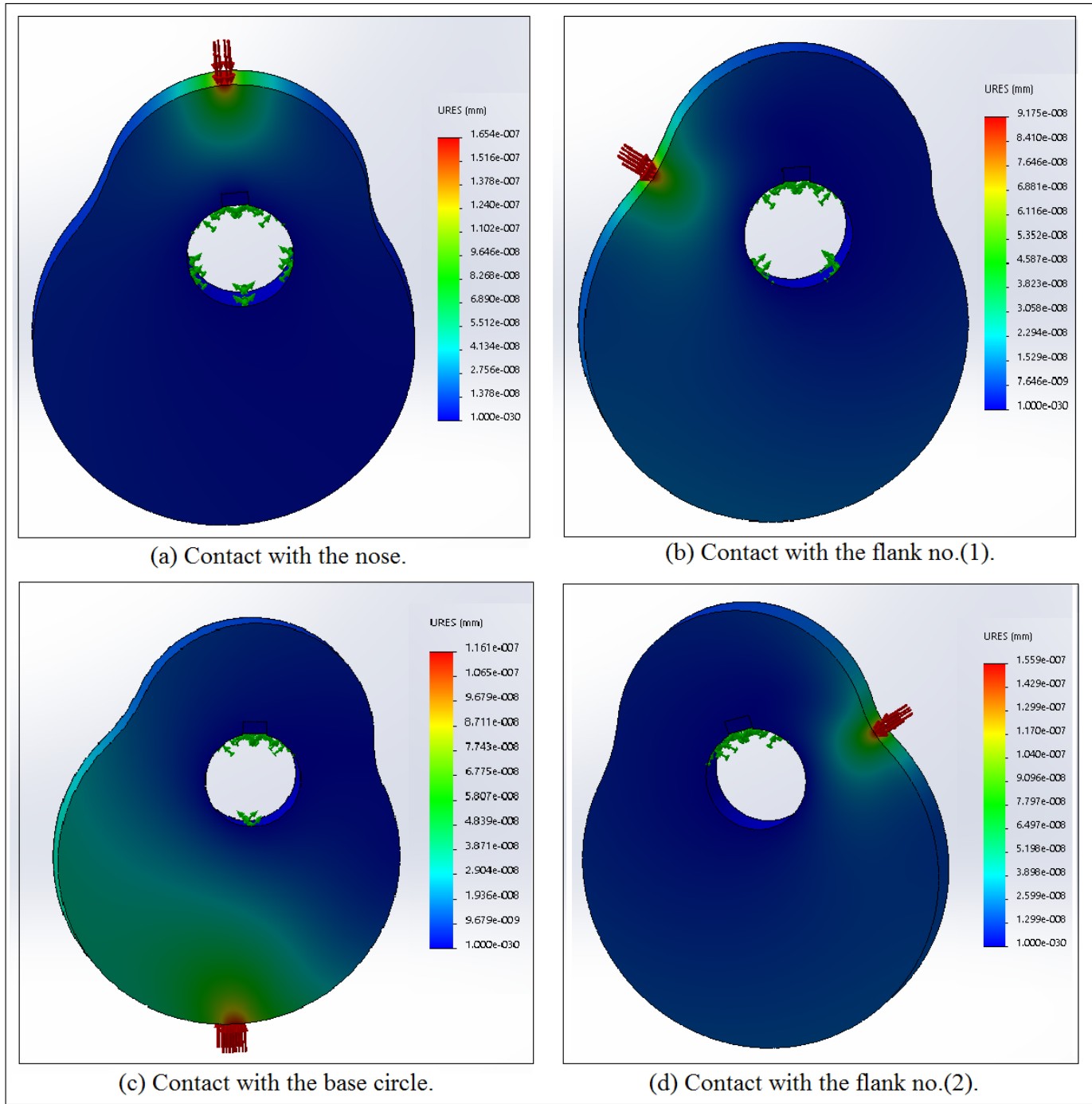

**Figure 21.** Bending deflection mapping contour for profile no.(2) at different contact locations.

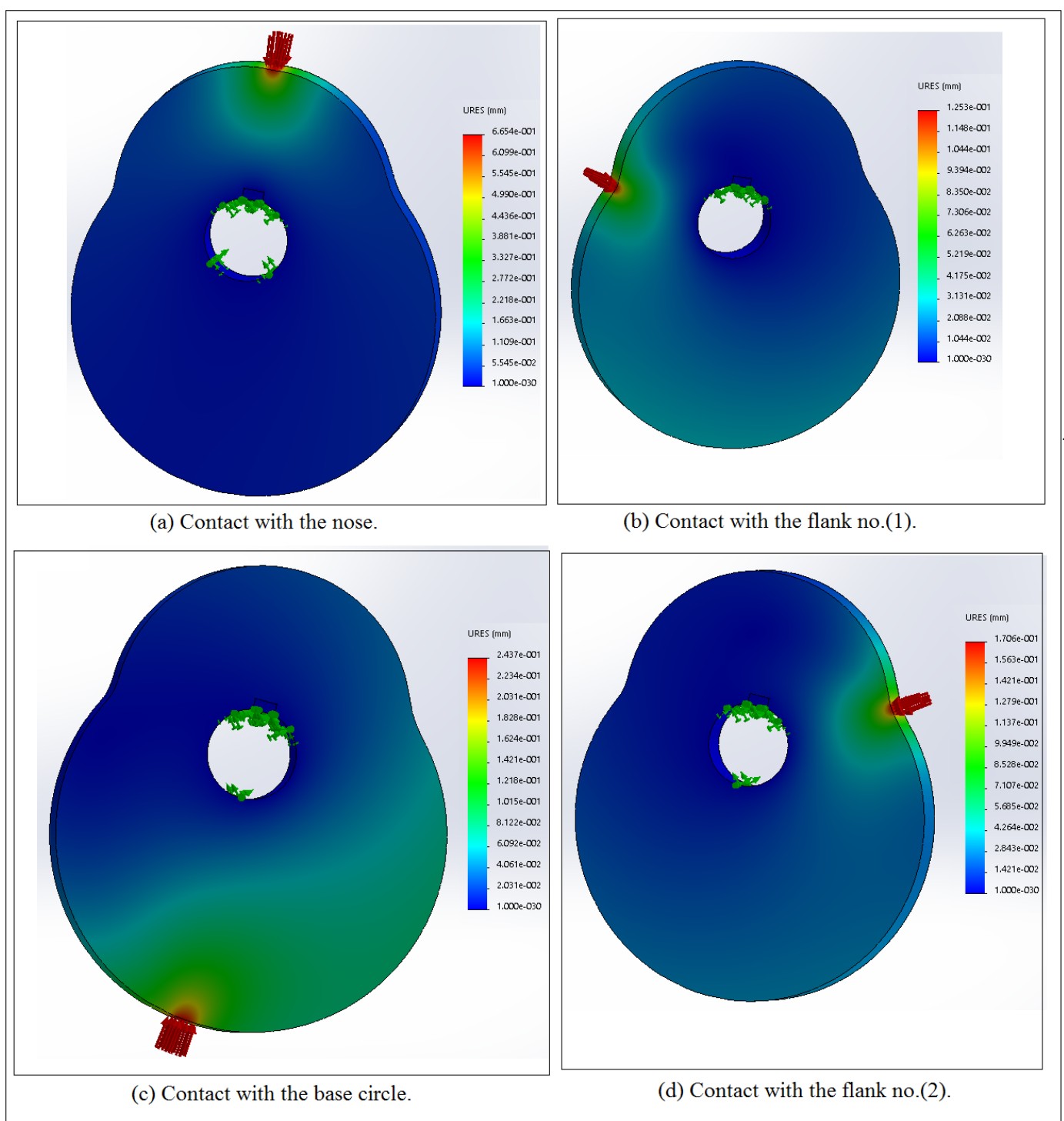

(a) Contact with the nose.

(b) Contact with the flank no.(1).

(c) Contact with the base circle.

(d) Contact with the flank no.(2).

**Figure 22.** Bending deflection mapping contour for profile no.(3) at different contact locations.

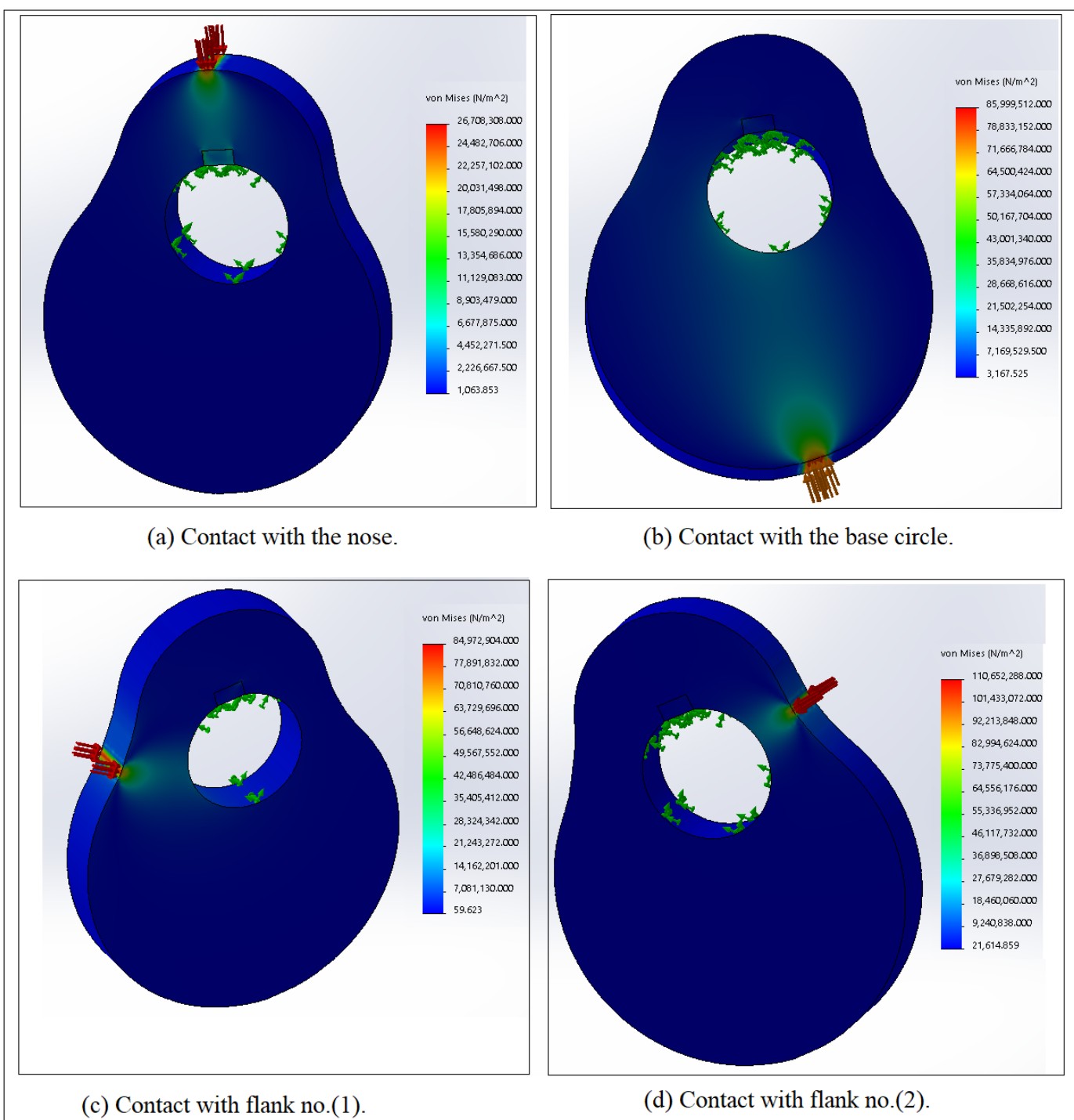

**Figure 23.** Von Mises stress mapping contour for profile no.(1) at different contact locations.

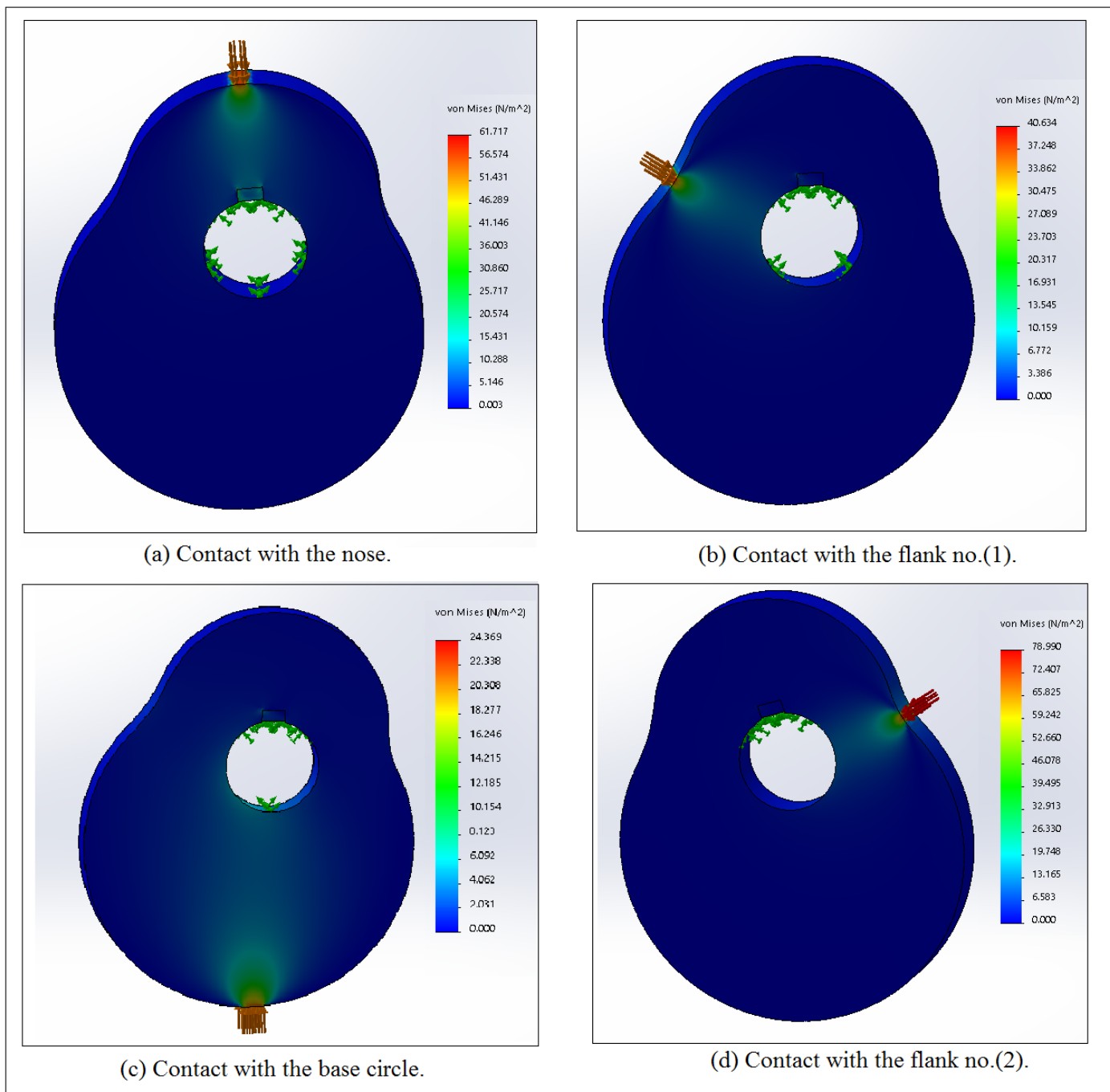

**Figure 24.** Von Mises stress mapping contour for profile no.(2) at different contact locations.

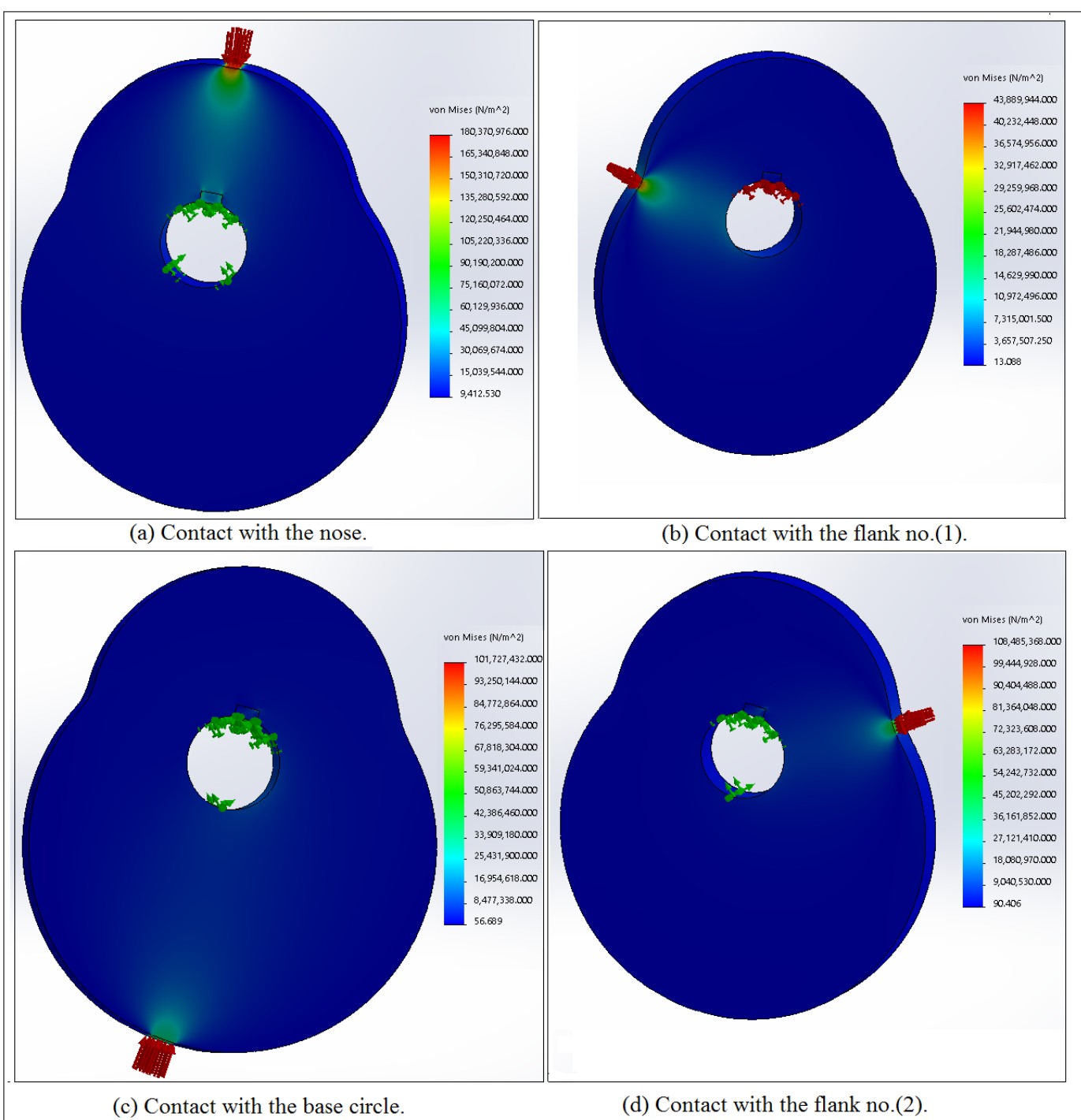

**Figure 25.** Von Mises stress mapping contour for profile no.(3) at different contact locations.

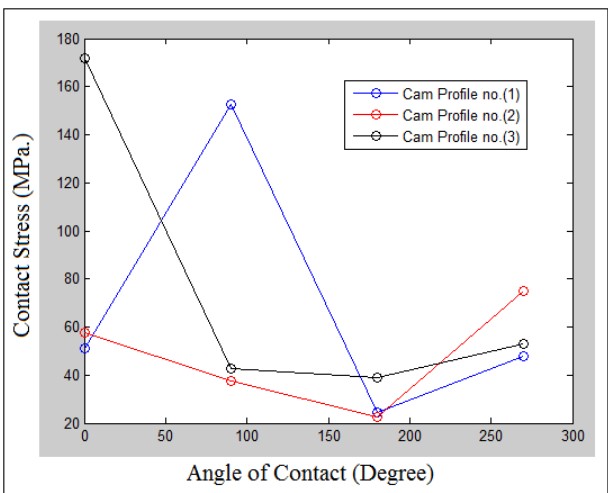

**Figure 26.** Contact stress against angle of contact at different involutes' profiles of the cam.

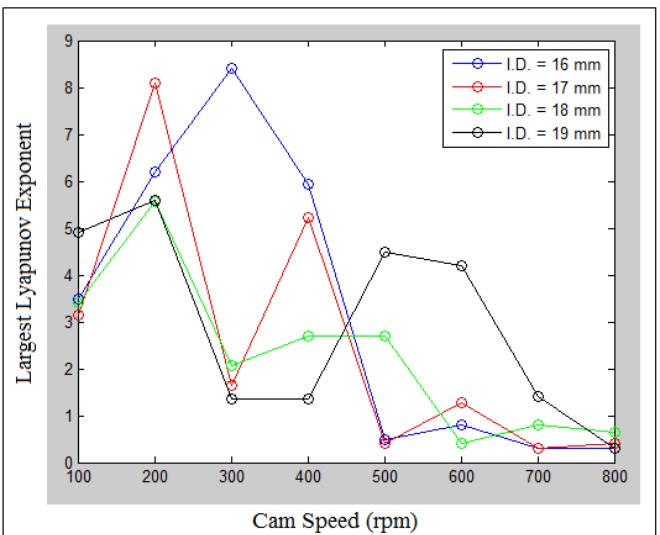

**Figure 27.** Largest Lyapunov exponent against cam speeds for cam profile no.(1).

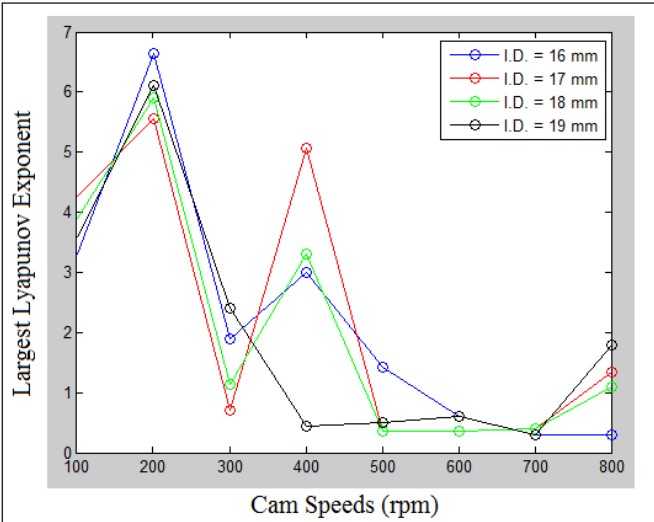

**Figure 28.** Largest Lyapunov exponent against cam speeds for cam profile no.(2).

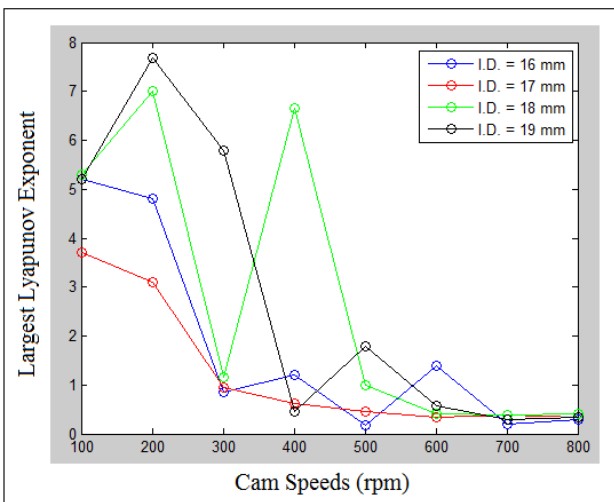

**Figure 29.** Largest Lyapunov exponent against cam speeds for cam profile no.(3).

**Figure 30.** Phase-plane mapping for cam profile no.(2) at different (I.D.) and different (N).

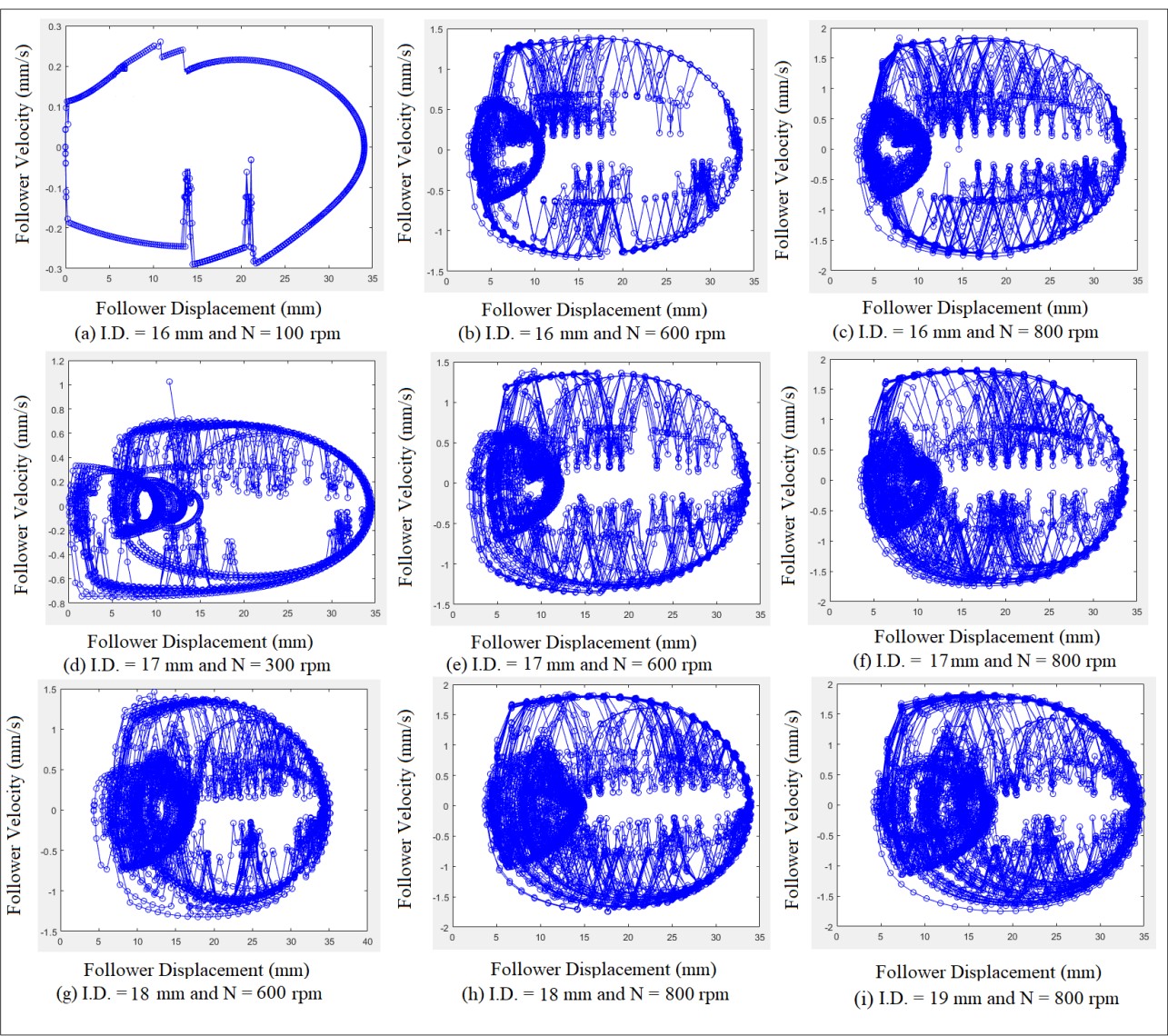

**Figure 31.** Phase-plane mapping for cam profile no.(3) at different (I.D.) and different (N).

## 9. Conclusions

The periodic and non-periodic motions of contact stress are investigated at different contact locations for a cam profile with a follower. The more the contact stress between the cam and the follower, the larger the largest Lyapunov exponent parameter. In the design, cam profile no.(2) can be selected because the bending deflection value is at the lowest level. The maximum value of the largest Lyapunov exponent is (LLE = 8.5) at (I.D. = 16 mm) and (N = 300 rpm) for cam profile no.(1), while the minimum value of the largest Lyapunov exponent is (LLE = 0.23) at (I.D. = 16 mm) and (N = 500 rpm) for cam profile no.(3). The cross-linking of the fringe order of the contact stress is increased with the increase in cam speeds (N) and with the increase in the internal distance of the follower guide from inside (I.D.). The value of the contact stress for the point of contact between the cam and the follower is higher than the value of the contact stress around the square grooving key since there is no dissipation in the potential energy. The more value of contact stress, the more value of bending deflection. Around the square grooving key, the contact stress has a maximum value for cam profile no.(3), while the contact stress has a minimum value for cam profile no.(2) at the nose location. The value of the contact stress at flank no.(2) is higher than the value of the contact stress at flank no.(1) because of the bigness of the radius of curvature of flank no.(2) for the three involutes' profiles of the cam. Regarding

the practical application, a three bending point test is used to extract the value of the stress fringe constant, which is needed in the main test of the photo-elastic device to calculate the stress distribution based on the fringe order. In general, the experiment part of the photo-elastic device is needed to investigate the value of the contact stress at the contact point and around the square grooving key. In the second part of the experiment setup, OPTOTRAK 30/20 device with a high-speed camera in the foreground is used to catch the follower displacement. Multi degrees of freedom on the follower stem are used to reduce the amplitude of vibration for the follower displacement and to keep the cam and the follower in permanent contact.

**Funding:** This article did not receive any kind of help and support from any department or institution.

**Institutional Review Board Statement:** Not applicable.

**Informed Consent Statement:** Not applicable.

**Data Availability Statement:** The data are available to everyone upon reasonable request.

**Conflicts of Interest:** The author declares that he has no conflict of interest.

## Nomenclature

| | |
|---|---|
| I.D. | Internal distance of the follower guide from inside, mm |
| N | Cam speed, rpm |
| I | Polar moment of inertia, $mm^4$ |
| $L_1$, $L_2$, $L_3$ | Dimensions of the follower stem, mm |
| $K_1$, $K_2$, $K_3$, $K_4$, $K_5$ | Spring stiffness, mm |
| $C_1$, C, $C_3$, $C_4$, $C_5$ | Damping coefficient, N·s/mm |
| $R_b$ | Radius of the base circle of the cam, mm |
| m | Mass of the follower, kg |
| $P_C$ | Contact force between the cam and the follower, N |
| Δ | Preload spring extension, mm |
| $\sigma_C$ | Contact stress between the cam and the follower, $N/mm^2$ |
| L | Thickness of the cam and the follower, mm |
| $R_1$, $R_2$ | Radius of curvature for the cam and the follower respectively, mm |
| d(t) | Rate of change in the distance between nearest neighbors |
| $d_j(i)$ | Distance between the jth pair at (i) nearest neighbors, mm |
| t | Single time series, s |
| y(i) | Curve fitting of least square method for the follower displacement data |
| Ø | Phase shift angle of the vibration mode shape, Degree |
| $Ø_1$ | Pressure angle, degree |
| λ | Lyapunov exponent |
| Ω | Eigenvalue problem, rpm |
| Δ(t) | Discrete time steps, s |
| D | Average displacement between trajectories at (t = 0) |
| N | Fringe order |
| Fs | Shear Force, N |
| h | Thickness of PMMA material, mm |
| $I_1$ | Second moment of inertia, $mm^4$ |
| y | Distance from neutral axis, mm |
| M | Bending moment of three bending point test, N.mm |
| $L_o$ | Distance of PMMA specimen of three bending point test, mm |
| $\sigma_1$, $\sigma_2$ | Principal stress in the direction 1, and 2 respectively |

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
