# Peer review of "Influence of Nonlinear Dynamics Behavior of the Roller Follower on the Contact Stress of Polydyne Cam Profile"

_processes, doi:10.3390/pr10030585_

Round 1
Reviewer 1 Report
The manuscript was significantly improved and mostly addressed the concerns. The revised result section greatly improved the completeness of the manuscript. The rewritten conclusion is better than before, but the discussion within is still very superficial. I suggest adding a few more sentences on the implications of your results on practical applications. This gives you a chance to emphasize why your results are needed. Also, I still see a number of mistakes and awkward sentences. I understand it is hard to be completely error-free, but I suggest trying to fix the language as much as possible.Author Response
I paraphrased the whole manuscript and I added new sentences regarding the practical application in the conclusion section.

Reviewer 2 Report
This time the paper has been substantially improved.
The conclusions section, has been improved.
It can now be published
Author Response
I paraphrased the whole manuscript and I added new sentences regarding the practical application in the conclusion section.

This manuscript is a resubmission of an earlier submission. The following is a list of the peer review reports and author responses from that submission.
Round 1
Reviewer 1 Report
Thanks for the interesting manuscript, I have a few remarks:
- Check the text, the size where the bold text should be removed.
- Compared to the size of the main text, the text size in the pictures should not be too large. Figures 2, 4, 2 needs higher resolution.
- Check the numbering and size of the equations.
Author Response
Response to Reviewer 1 Comments
Point 1:
Check the text, the size where the bold text should be removed.
Response 1:
The bold text in the manuscript has been removed.
Point 2:
Compared to the size of the main text, the text size in the pictures should not be too large. Figures 2, 4, 2 needs higher resolution.
Response 2:
Figures 2 and 4 has been removed and the other figures that have test size too large are fixed. Adobe Photoshop CS6 is used to increase the resolution of the figures.
Point 3:
Check the numbering and size of the equations.
Response 3:
The number and size of the equations has been fixed.
Note: The introduction section is paraphrased by adding some references and by deleting the unnecessary references.

Reviewer 2 Report
The manuscript addresses the effect of nonlinear dynamics behavior of roller follower of bionic quadruped robots on the stress concenctration at different cam locations. Thereby, experimental and numerical methods were employed. It was reported, that largest Lyapunov exponent is proportional to the contact stresses. My evaluation is as follows:
First of all, the main function of the cam is a transmission of information/motion. The profile cannot be adjusted arbitrarily to reduce the stress. This certainly holds true for bionic quadruped robots. Therefore, I question the meaningfulness of the investigation.
In addition, the manuscript has some serious weaknesses:
- The abstract is hard to follow as it is very specific. For example, the reader has no clue what profile no. (2) and (3) are at his point. Moreover, less information on the used methods (especially not the type of equipment) is required but information on the insights would be desired. Also, abbreviations are introduced that are not used in the abstract.
- The research gap to be filled by this article remains unclear.
- The overall length of the manuscript is too long (44 pages). It should be written more concisely and end up at no more than 15-30 pages (in MDPI format). The author should focus on the most important results and try to summarize them in a profitable manner. Additional information could be provided as a Supplementary File, for example.
- There are too many figures (44!), many of which are unnecessary and of poor quality. Figure 6 even appears to be scanned from a textbook? A research article shoudn't contain more than 15 items (tables and figures).
- The manuscript should be split into a theory/materials/methods section, a results section, and a discussion section. The chronological description somewhat hides it, but a proper discussion of the results - possibly in the context of the literature - is missing after all.
- The results only bring few practical implications. The main focus of the article is on the methods and a set of results. However, the latter are insufficiently interpreted and generalized.
Overall, the article has inherent weaknesses in terms of content and structure. Therefore, I clearly recommend a rejection.
Author Response
Response to Reviewer 2 Comments
Point 1:
The main function of the cam is a transmission of information/motion. The profile cannot be adjusted arbitrarily to reduce the stress. This certainly holds true for bionic quadruped robots.
Response 1:
The main function of this study is the use of short data of the contact stress against time around the nose, flank 1, base circle, and flank 2 in wolf algorithm code to quantify the value of largest Lyapunov exponent parameter. From Figure 10 it can be seen that the contact stress is maximum at the nose location in which the more value of contact stress at the nose location of the cam profile, the more value of Lyapunov exponent. The contact stress at the nose is (180 MPa.) which gives indication to non-periodic motion and chaos while the contact stress at the flank 1 is (43 MPa.) which means that the motion of the follower it could be either periodic or quasi-periodic because of the small value of radius of curvature. Three types of cam profile are taken into account to see which cam profile gives small value of contact stress and small value of largest Lyapunov exponent.
Point 2:
The abstract is hard to follow as it is very specific. For example, the reader has no clue what profile no. (2) and (3) are at his point. Moreover, less information on the used methods (especially not the type of equipment) is required but information on the insights would be desired. Also, abbreviations are introduced that are not used in the abstract.
Response 2:
The abstract section is written again and I put a specific details about the theory, material, method and conclusion. The abbreviation have been removed. Please check the new version of the manuscript.
Point 3:
The research gap to be filled by this article remains unclear.
Response 3:
Yes the gap is filled by studying the relationship between the contact stress and largest Lyapunov exponent parameter. The contact stress distribution has a direct effect on the nonlinear dynamics phenomenon of the follower. Equation (14) shows that the contact stress is a function of radius of curvature (R1) and (R2) at the point of contact between the cam and the follower. (R1) could be convex radius of the nose, concave radius of flank 1, convex radius of the base circle, or concave radius of flank 2 but (R2) should be radius of the roller follower. Reference [1] studied the effect of concave radius of curvature of the cam profile on the nonlinear dynamic phenomenon of the follower at different cam speeds and different internal distance of the follower guide from inside. If the contact stress is a function of radius of curvatures based on equation (14) and the radius of curvatures affect the nonlinear dynamics phenomenon of the follower, therefore the contact stress between the cam and the follower will affect the nonlinear dynamics phenomenon of the follower by substitution.
Point 4:
The overall length of the manuscript is too long (44 pages). It should be written more concisely and end up at no more than 15-30 pages (in MDPI format). The author should focus on the most important results and try to summarize them in a profitable manner. Additional information could be provided as a Supplementary File, for example.
Response 4:
I reduced the number of pages to 17 pages.
Point 5:
There are too many figures (44!), many of which are unnecessary and of poor quality. Figure 6 even appears to be scanned from a textbook? A research article shoudn't contain more than 15 items (tables and figures).
Response 5:
All the unnecessary figures have been deleted and I kept just 15 figures. I used Adobe-Photoshop CS6 to increase the resolution of these figures.
Point 6:
The manuscript should be split into a theory/materials/methods section, a results section, and a discussion section. The chronological description somewhat hides it, but a proper discussion of the results - possibly in the context of the literature - is missing after all.
Response 6:
The manuscript is divided to: Theory of vibration of analytic nonlinear response of the follower section, Numerical simulation section, Finite element analysis steps using SolidWorks program section, Largest Lyapunov exponent parameter section, Methods of experiment setup section, Results section, discussion section, and conclusion section.
Point 7:
The results only bring few practical implications. The main focus of the article is on the methods and a set of results. However, the latter are insufficiently interpreted and generalized.
Response 7:
I deleted all the results that do not reflect any relationship between the contact stress and largest Lyapunov exponent parameter. I focused on the results that give indication to periodic and non-periodic motion and chaos based on the set of data of contact stress against time using the conception of largest Lyapunov exponent parameter in the discussion section.
Note:
The English style and spell are checked and corrected in the new version of the manuscript. The introduction section is paraphrased by adding some references and by deleting the unnecessary references.

Reviewer 3 Report
Please answer the following:
- consider the changes highlighted in yellow in the resend paper;
- please explain where you used the analytical calculations in Chapter 2;
- please explain why you did not use, in the analysis with finite elements, a differentiated network that must be more dense in contact area;
- are finite element analysis programs that provide better contact results than Solid Works;
- as can be seen in you paper (in the second part of relation 7), the experiment provides diferences in stresss, s1-s2; how in table 1 the comparison with the stresses from finite elements is made?; As can be seen from Figure 10, the stresses in finite elements are von Mises.

Reviewer 4 Report
This manuscript focuses on explored the contact mechanics for a cam follower system; more specifically, the author investigated the effect of nonlinear dynamics on the contact stress between the cam and the follower at high speeds with the goal to suppress the nonlinear dynamics phenomenon of the follower and to reduce the contact stress between the cam and the follower. To do this, the author use experiments and simulations to test 3 cam profiles to find the optimal shape for the cam.
There are numerous mistakes throughout the paper, lack details in many areas of the paper, the language used throughout the paper is awkward, and there are no text descriptions of the results. The paper is poorly written; it is in a very poor and unfinished state. I recommend resubmission after significant revisions.
- The manuscript has a lot of mistakes. Please proofread.
- The is a comma in front of every in-text citation.
- The results from Figure 7 seems very similar to the results from previous papers. Why repeat such analysis for this work when it was already done previously? If they are different, please elaborate to avoid confusion.
- In line 203, what are “numbers (1), (2), (3)”? Do you mean profiles?
- The results section has no text, only figures. Please add text.\
- How come only profile 1 was present in figure 10 and figure 15?
- The conclusion is written like it is a description of the results. This is not what a conclusion should be. Give conclusions in the conclusion section. Describe your results in the results section.
Round 2
Reviewer 2 Report
The author has put some efforts into revising the manuscript. However, in my optinion, the paper still has too many weaknesses and the novelty and generalizability of the findings are not sufficient for publication. I stand by my original recommendation of rejection.